# PLAN-AND-ACT: Improving Planning of Agents for Long-Horizon Tasks

**Lutfi Eren Erdogan** [* 1]  **Nicholas Lee** [* 1]  **Sehoon Kim** [1]  **Suhong Moon** [1]  **Hiroki Furuta** [2]
**Gopala Anumanchipalli** [1]  **Kurt Keutzer** [1]  **Amir Gholami** [1 3]

## Abstract

Large language models (LLMs) have shown remarkable advancements in enabling language agents to tackle simple tasks. However, applying them for complex, multi-step, long-horizon tasks remains a challenge. Recent work have found success by separating high-level planning from low-level execution, which enables the model to effectively balance high-level planning objectives and low-level execution details. However, generating accurate plans remains difficult since LLMs are not inherently trained for this task. To address this, we propose PLAN-AND-ACT, a novel framework that incorporates explicit planning into LLM-based agents and introduces a scalable method to enhance plan generation through a novel synthetic data generation method. PLAN-AND-ACT consists of a PLANNER model which generates structured, high-level plans to achieve user goals, and an EXECUTOR model that translates these plans into environment-specific actions. To train the PLANNER effectively, we introduce a synthetic data generation method that annotates ground-truth trajectories with feasible plans, augmented with diverse and extensive examples to enhance generalization. We evaluate PLAN-AND-ACT using web navigation as a representative long-horizon planning environment, demonstrating a state-of-the-art 57.58% success rate on the WebArena-Lite benchmark as well as a text-only state-of-the-art 81.36% success rate on WebVoyager.

## 1. Introduction

Large language models (LLMs) have significantly advanced in capability, enabling their application as *language agents*

---

[*]Equal contribution  [1]UC Berkeley  [2]University of Tokyo  [3]ICSI. Correspondence to: Amir Gholami <amirgh@berkeley.edu>.

*Proceedings of the 42nd International Conference on Machine Learning*, Vancouver, Canada. PMLR 267, 2025. Copyright 2025 by the author(s).

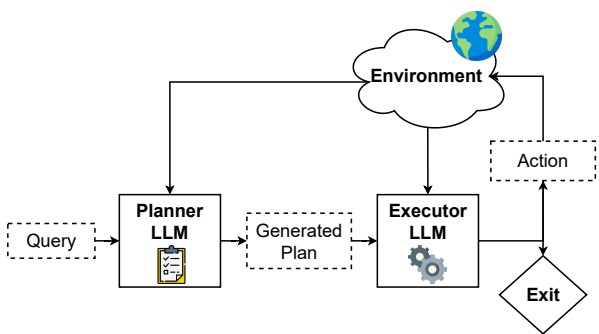

*Figure 1.* An illustration of PLAN-AND-ACT System Diagram. First, the PLANNER LLM processes the initial user query and generates an initial step by step plan (Section 3.1). This is then passed to the EXECUTOR LLM which uses the plan and generates an actions to interact with its Environment. The environment feedback is then fed back to both the EXECUTOR so it can generate subsequent actions and/or to the PLANNER in case a new plan needs to be generated. Existing methods have shown this separation of high-level planning and low-level execution can improve accuracy. However, a major challenge is that LLMs are not generally trained to generate such plan/low-level action, a problem that we focus on solving in this paper.

that can interact with environments through sequences of actions. These agents are designed to tackle complex, multi-step, long-horizon tasks by leveraging the model's reasoning and decision-making capabilities. At the heart of building such effective agents lies a fundamental challenge: planning. Even for seemingly simple tasks, an agent must understand the goal, break it down into manageable steps, and adapt those steps as circumstances change. However, despite these advancements, planning remains a significant challenge for several reasons. First, agents often struggle to break down high-level user goals (like "book me a flight to New York") into specific, actionable steps (like "open the airline website", "enter travel dates", etc.). Second, as tasks grow longer and more complex, maintaining a coherent strategy becomes increasingly difficult - agents lose track of what they've accomplished and what remains to be done. Third, real-world environments are dynamic and unpredictable, requiring agents to constantly revise their plans. These challenges are further amplified by the scarcity of high-quality

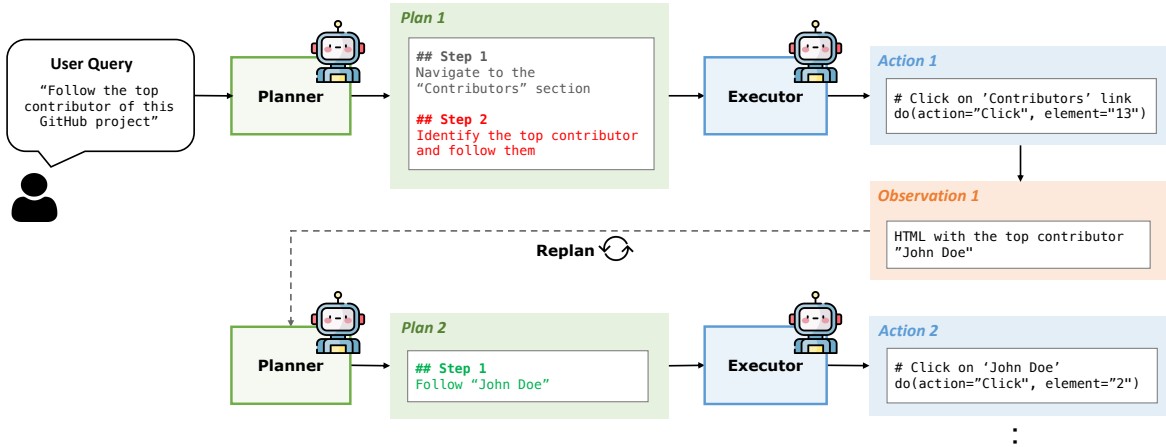

*Figure 2.* PLAN-AND-ACT System Diagram. Given the initial user query, the PLANNER (Section 3.1) breaks it down into a high-level plan, which is given to the EXECUTOR (Section 3.2) which uses the plan to guide its actions. Once the action has been taken and the HTML changes, the PLANNER dynamically generates a new plan that incorporates the changes in the environment (Section 3.3).

training data that demonstrate effective planning strategies.

Previous approaches to improve agent performance for long-range tasks such as web navigation (Qi et al., 2024; Yang et al., 2024a; Gur et al., 2023; Zhang et al., 2024b) and device control (Erdogan et al., 2024; Pawlowski et al., 2024) have shown promise. However, most rely on a single model to directly translate user requests into actions. This creates a difficult balancing act - the model must simultaneously reason about the high-level strategy while managing the low-level execution details (Yang et al., 2024a). Under this load, models often lose sight of their ultimate objectives and struggle to maintain consistent behavior (Sridhar et al., 2023). Recent work has explored the use of Reinforcement Learning (RL) (Sutton, 2018) to improve performance (Bai et al., 2024; Qi et al., 2024), but these methods can be unstable and highly sensitive to hyperparameters and reward design (Rafailov et al., 2024; Furuta et al., 2024).

To equip LLMs to effectively plan for long-horizon tasks more reliably, we introduce PLAN-AND-ACT, a framework that incorporates explicit planning into LLM-based agents. Unlike traditional approaches like ReAct based methods (Qi et al., 2024; Gur et al., 2023; Yang et al., 2024b) that rely on a single model to directly map user queries to a sequence of actions, PLAN-AND-ACT adopts a framework similar to LLMCompiler (Kim et al., 2023b) which consists of two modules: a PLANNER and an EXECUTOR (Figure 1). The PLANNER model generates a sequence of plans that outline the high-level steps required to achieve the goal, while the Executor translates these steps into environment-specific actions. Furthermore, our framework provides an effective and scalable solution for generating training data to train the PLANNER without requiring manual annotation or a sandbox environment. The difference with LLMCompiler is

that we introduce a scalable method that allows finetuning the PLANNER and EXECUTOR components. In particular, our contributions are as follows:

- We propose PLAN-AND-ACT, a framework that improves planning for long-horizon tasks through explicit separation of planning and execution. As shown in Figure 2, our architecture consists of a PLANNER that breaks down user requests into structured plans, and an EXECUTOR that implements these plans through environment-specific actions (Section 3).

- To train the PLANNER model effectively, we introduce a synthetic data generation pipeline to generate planner data with and without access to extra ground truth data. First, we use an LLM to analyze successful action trajectories (sequences of actions like clicking, typing, etc.) and generate the corresponding high-level plans through grounded plan generation, ensuring these plans are grounded in actual executable actions (Section 4.2). Second, we synthetically augment our dataset by using these initial plans as seed data to generate additional diverse planning examples (Section 4.3). This comprehensive approach enables us to create high-quality training data despite the scarcity of real-world planning examples.

- To demonstrate the efficacy of our approach on long-horizon tasks, we evaluate PLAN-AND-ACT in the WebArena-Lite (Liu et al., 2024) environment for web navigation, achieving SOTA result of 53.94% (Table 1).

## 2. Related Work

### 2.1. Language Agents as Web Agents

As LLM agents become more widespread, they have been increasingly applied as web agents to traverse and operate web pages (Gur et al., 2022; Kim et al., 2023a) through GUI interaction (Pawlowski et al., 2024; Rawles et al., 2024; Zhang et al., 2023; Xie et al., 2024; Zhang et al., 2024a) or API interaction (Song et al., 2024), which has spurred new datasets (Shi et al., 2017; Deng et al., 2024; Rawles et al., 2024) and benchmarks (Yao et al., 2022a; Zhou et al., 2023; Koh et al., 2024; Liu et al., 2024).

Several works employ hierarchical planning frameworks similar to PLAN-AND-ACT, but rely on prompting closed-source models like GPT-4o. AgentOccam (Yang et al., 2024a) incorporates planning into the action space with tree-like planning, WebPilot (Zhang et al., 2024b) uses six different agents, AdaPlanner (Sun et al., 2023) employs In-Plan and Out-of-Plan Refiners for replanning, and ADaPT (Prasad et al., 2023) uses recursive decomposition. In contrast, PLAN-AND-ACT provides a simpler two-agent framework with a systematic approach to generating high-quality training data for open-source LLMs.

Training data generation approaches for Web Agents include DigiRL (Bai et al., 2024), WebRL (Qi et al., 2024), AutoWebGLM (Lai et al., 2024), and NNetNav (Murty et al., 2024), which focus on collecting diverse trajectories but lack planning data generation. Unlike these methods that rely on external simulators, our approach can generate synthetic planning data without a simulator (Section 4.3).

Some robotics research (Song et al., 2023; Nayak et al., 2024; Kannan et al., 2024) uses hierarchical LLM Agents for task decomposition and planning, sharing similarities with our work. Other approaches include pretraining LLMs on HTML (Gur et al., 2023), leveraging vision capabilities of VLMs (Furuta et al., 2023), or adopting RL to improve performance through interaction (Qi et al., 2024; Bai et al., 2024; Yang et al., 2024b).

### 2.2. Synthetic Data Generation

Synthetic generation has gained popularity since pioneering work like Self-Instruct (Wang et al., 2022) and Alpaca (Taori et al., 2023), with many recent papers using synthetic data to enhance LLM performance (Xu et al., 2023; Lee et al., 2024; Erdogan et al., 2024; Wang et al., 2024a; Gunasekar et al., 2023; Moon et al., 2024).

For web agents specifically, researchers commonly collect training trajectories in reusable environments (Zhou et al., 2023; Koh et al., 2024; Liu et al., 2024) through on-policy methods (Qi et al., 2024; Yang et al., 2024b; Patel et al., 2024; Ou et al., 2024). These approaches typically generate trajectories from an LLM and filter for failed instances. Patel et al. (2024) combined real and synthetic data, while Yang et al. (2024b) supplemented training with multiple trajectories on failed tasks. Both NNetscape Navigator (Murty et al., 2024) and WebRL (Qi et al., 2024) employed instruction-generation techniques—the former retroactively labeled explored website trajectories, while the latter used Self-Instruct style prompts to generate queries for trajectory collection, with failed trajectories seeding further targeted data generation.

## 3. System Architecture

As discussed in previous work (Sridhar et al., 2023; Yang et al., 2024a), at the heart of effective agents lies the challenge of balancing high-level reasoning with low-level execution. When a single model must simultaneously perform long horizon planning and then also execute multiple low-level actions for each part of the plan, it faces a difficult cognitive load that often leads to suboptimal decisions or inconsistent behavior. This challenge becomes especially acute for long-horizon tasks, where the agent must maintain a coherent strategy across many steps while adapting to changes in the environment.

To address this fundamental challenge, our framework separates these responsibilities into two specialized components: a PLANNER that focuses on strategic decision-making and an EXECUTOR that specializes in implementing those decisions (Figure 2). This separation allows each component to excel at its core task. The PLANNER can reason about high-level strategy without getting bogged down in implementation details, while the EXECUTOR can focus on translating abstract plans into concrete actions (Kim et al., 2023b; Wang et al., 2023).

While our framework is adaptable to various structured decision-making environments, we focus on web agents due to the web's dynamic and complex nature, which involves diverse actions and long-horizon tasks. For web tasks, the PLANNER takes a user query (like "Follow the top contributor of this GitHub project") and breaks it down into clear, manageable steps (such as "Navigate to the Contributors section" followed by "Identify and follow the top contributor"). The EXECUTOR then takes these steps and translates them into precise actions in the web environment, like clicking specific links or typing in search boxes. The observation space that we will use for this task is HTML as the text-representation of the environment.

### 3.1. PLANNER

The PLANNER takes the user query and breaks it down into a structured plan that dictates the essential high level steps required to accomplish the task. This plan is used to as a

guide for the EXECUTOR at runtime, providing a clear roadmap while allowing some flexibility for the EXECUTOR. By handling most of the reasoning and task decomposition, the PLANNER streamlines decision making and task execution.

Consider the example in Figure 2, which shows an example of this module in action, in the context of a web task. We have the original user query *Follow the top contributor of this GitHub project* and the goal is for our WebAgent to execute this task on GitHub. The PLANNER processes this user query and breaks down the task into a step-by-step plan, consisting of (i) Navigating to the Contributors section and (ii) Identifying and Following the top contributor.

### 3.2. EXECUTOR

The EXECUTOR is an LLM Agent that takes the plan from the PLANNER (Section 3.1) and runs it in the environment. It is responsible for calling tools, retrieving data, or making changes in the environment required by the plan.

For example, in the web task shown in Figure 2, once the PLANNER generates a plan for the query *Follow the top contributor of this GitHub project*, the EXECUTOR only needs to translate the step *Navigate to the "Contributors" section* into a click action on the HTML. As we can see, EXECUTOR takes HTML as an input and outputs a grounded concrete action in the environment. Importantly, after executing an action, the EXECUTOR performs garbage collection, removing unnecessary data such as redundant HTML before executing the next action. More explicit examples of the PLANNER-EXECUTOR trajectories can be found in Appendix A.4.

### 3.3. Dynamic Replanning

A key limitation of the previous approach is that the initial plan is *static* throughout execution, which makes it vulnerable to unexpected variations in the environment. For instance, in the web-navigation example, static plans are unequipped to handle dynamic content interpretation, such as analyzing search results or transaction histories. The EXECUTOR can fail to correctly process content that is unknown a priori at planning time. Furthermore, static plans can have issues with unexpected failures, such as searching a keyword returning nothing. If the plans are static, the EXECUTOR may blindly follow the steps in the original plan rather than trying a different approach. As shown in previous work in task decomposition (Kim et al., 2023b), static planning can have fundamental drawbacks even in straightforward tasks.

To address this limitation, we introduce *dynamic replanning*, where the PLANNER updates the plan after each EXECUTOR step rather than relying solely on the initial plan. After each iteration, the PLANNER takes in the *current* state as well as the previous plans and actions and generates a new plan for

how the EXECUTOR can complete the user query.

Conveniently, dynamic replanning allows the planner to retain key information within the evolving plan. For instance, consider the example in Figure 2. The original plan did not know who the top contributor was, so it could only contain the step *Identify the top contributor and follow them*. After the contributor was identified upon execution of the action *clicking "Contributors" link*, the PLANNER incorporates this information into the remaining plan. Since the plan carries forward the relevant context, this approach also allows us to address challenges related to memory for long-horizon tasks without requiring an explicit memory module (Yang et al., 2024a; Zhang et al., 2024b). More detailed examples of dynamic replanning can be found at Appendix A.5.

This approach aligns with our architectural philosophy where the PLANNER serves as the "control room" for reasoning and decision-making, while the EXECUTOR focuses solely on translating plans into environment-specific actions.

### 3.4. Chain of Thought Reasoning

Currently, the PLANNER and EXECUTOR generate plans and actions directly. However, recent advances in chain-of-thought (CoT) prompting and inference-time scaling (Kojima et al., 2022; Wei et al., 2022; Guo et al., 2025) have shown that eliciting intermediate, step-by-step rationales can substantially improve performance. Thus, before having the PLANNER and EXECUTOR generate the plan and action respectively, we also have them generate a CoT reasoning trace in order to improve performance. You can find the CoT experiment results in Section A.1

## 4. Synthetic Data Generation

To motivate the need for creating synthetic data, we first evaluated the performance of existing off-the-shelf LLMs on WebArena-Lite which involves challenging user queries and reported the results in Table 1. We observe a baseline performance of 9.85%, which increases to 14.21% with PLAN-AND-ACT. While this is a noticeable improvement, the result is far from satisfactory.

There are several contributing factors to this low performance. Most notably, LLMs are not trained to perform this form of long horizon planning, especially for web tasks. This affects both the PLANNER and EXECUTOR. The PLANNER cannot generate accurate plans if the LLM has not seen these websites in its pretraining and the trajectories that are needed to accomplish the query. The EXECUTOR has also most likely not been trained on getting a user query along with the HTML of a page and output a web action. Overall, we cannot expect an off-the-shelf LLM to have this capability if it has not been trained on planning/executing tasks during pretraining such as in a specialized domain, such as

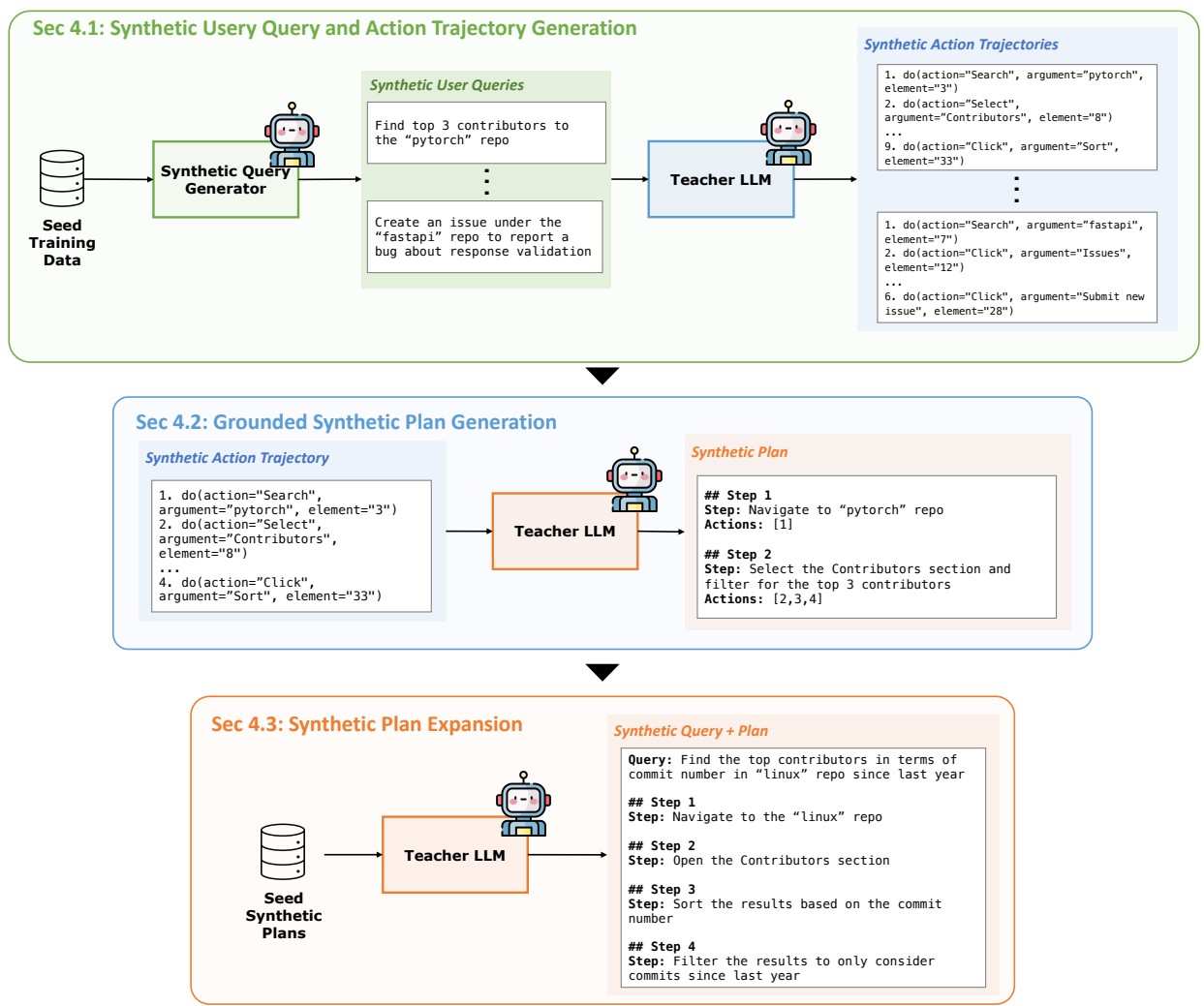

*Figure 3.* Synthetic Data Generation Pipeline. In the Action Trajectory Generation stage (Section 4.1), user queries from the training data are given to a Teacher LLM, which outputs synthetic user queries. From there, a demonstrator actor LLM attempts to execute the query on the webpage. After the trajectory is finished, an ORM LLM is used to filter for successful trajectories. In the Grounded Plan Generation stage (Section 4.2), a Teacher LLM takes the trajectory and creates a synthetic high-level plan and grounds each step with explicit actions in the trajectory. In the Synthetic Plan Expansion stage (Section 4.3), the plans from the training data are sampled and given to the Teacher LLM, which generates new synthetic plans.

web navigation.

Importantly this issue cannot be solved with prompting alone. While performing prompt engineering and including in-context examples can help with simple tasks, the LLM struggles when given non-trivial queries as evident by the low baseline accuracy.

Given that prompting alone cannot solve this issue, we have to perform finetuning of the LLM to improve the performance of the PLANNER and EXECUTOR. However, in order to finetune the model, we need to ensure that we have sufficient amount of data for finetuning. In particular, the PLANNER requires data that has a user query and its corre-

sponding plan breakdown. The EXECUTOR needs data that includes HTML input for each of the plan steps along with the desired web action as its output. However, such data is not available. Manually collecting this data is an option but is both costly and time-consuming as the human annotator needs to write down a plan of action followed by execution corresponding to each of the steps.

Here we propose an alternative approach that allows a scalable method to collect and generate high-quality synthetic training data. Our method leverages an LLM-based annotation pipeline that processes existing action trajectories to generate the corresponding structured plans, which is depicted in Figure 3.

## 4.1. Action Trajectory Generation

The simplest way to collect more data for the EXECUTOR is to collect more trajectories from the environment, a technique that has been used in previous works (Qi et al., 2024; Murty et al., 2024).

To achieve this, we use an Alpaca-style (Taori et al., 2023) data generation pipeline. Motivated by Qi et al. (2024), we randomly sample user queries from the training data and use them as seed prompts for an LLM to generate new, similar queries. For the web-navigation task, these queries are filtered initially by an LLM to filter out impossible trajectories for the web-agent. These newly generated user queries are then given to a demonstrator agent which tries to complete the task in the environment, which we collect as a synthetic trajectory. Finally, we then score this trajectory by using an outcome-supervised reward model (ORM) to filter for successful and unsuccessful trajectories. This initial process is depicted in Figure 3 in the first row for the web-navigation task and the prompts for the query generation are adapted from (Qi et al., 2024).

## 4.2. Grounded Plan Generation

While we can use synthetically created user queries and then collect the resulting trajectories to train the EXECUTOR, this approach presents challenges when used to create synthetic data for the PLANNER.

A naive approach would be to provide a teacher LLM with a user query and prompt it to generate a step-by-step plan to accomplish the task. However, this runs into a fundamental limitation: the teacher LLM lacks access to the actual website or environment and has not been pretrained on such tasks. Attempting this method results in generated synthetic plans that are often misaligned with how the tasks need to be performed on the web.

To address this, we leverage the in-context learning capabilities of LLMs, which allows them to generalize on tasks outside their pretraining distribution. Specifically, we take advantage of this capability and provide the teacher LLM the trajectories that we created in Sec. 4.1 and prompt it to "reverse-engineer" structured plans from these trajectories. Given the trajectory, we prompt the LLM to analyze the sequence of actions and to synthesize a coherent plan that will be used to guide the EXECUTOR downstream. To make sure that the plan is grounded to the actual environment, we further prompt the model to also include which low-level actions in the trajectory would be assigned to which high-level actions in the plan, to ensure that the plan matches actual execution of the trajectory. This ensures that the generated plans align with the real execution environment, making the both accurate and executable.

This is depicted in the second row of Figure 3, where the action trajectories of the webagent is transformed into a set of high-level actions that we want the PLANNER to output. This approach is similar to (Murty et al., 2024), although our method generates high-level plans while their technique generates synthetic user queries from the trajectories.

### 4.2.1. SYNTHETIC DATA GENERATION FOR DYNAMIC REPLANNING

It is important to also create synthetic data that captures dynamic replanning. This is important because a lot of user queries require planning based on dynamic observations that are only known during the plan execution. Examples queries that require such planning are: *"Analyze the search results and select the most relevant item"* or *"Find the most recent pending order"*.

We can use a similar algorithm to generate synthetic replanning data. The main difference is that for replanning data generation, we need to supply the teacher LLM with original plan data along with the trajectory that the webagent has taken to reach the point that requires replanning. You can find detailed prompts in Appendix A.12.

### 4.2.2. SYNTHETIC DATA GENERATION FOR CHAIN-OF-THOUGHT-REASONING

Similarly, we also need to generate synthetic data to elicit CoT reasoning for both the PLANNER and EXECUTOR, since not all models have been trained to generate reasoning traces. We use an algorithm similar to Section 4.2 to generate reasoning traces for both plan and action generation. For plan reasoning generation, we have the teacher LLM generate reasoning before outputting the plan while for action reasoning generation, we provide the original plan data and the trajectory that the webagent has taken, along with the expected correct action and prompt the teacher LLM to generate a reasoning trace for that action.

## 4.3. Synthetic Plan Expansion

The previous approach requires a simulator environment to collect actions and then create a synthetic plan by reverse-engineering the actions. Collecting successful trajectories that passes the ORM in Sec. 4.1 can be time consuming since the teacher model may generate a lot of unacceptable trajectories. This will affect the amount of data that we can generate both for the EXECUTOR as well as the PLANNER. This issue is noticeably worse for the PLANNER since each successful trajectory that passes the ORM model entails on average 8 different steps which provide 8 training data points for the EXECUTOR, but only 1 plan. However, we can effectively address this data imbalance, by expanding the synthetic plans.

Specifically, we expand the PLANNER dataset by generating

similar query-plan pairs that resemble the existing data, similar to the Alpaca style query generation in Section 4.1.

Similar to Section 4.1, we initially randomly sample query-plan pairs from the synthetic PLANNER training data. These examples serve as implicit constraints, guiding the language model to generate structurally consistent and semantically valid query-plan pairs while maintaining diversity.

Using this pipeline with GPT-4o, we expanded the synthetic plan data to 10,000 additional user query-plan pairs. This approach demonstrated significant advantages in both efficiency and scalability, reducing data generation time to under an hour while simultaneously addressing the overfitting problem through increased data diversity. The resulting synthetic dataset exhibits a broad spectrum of use cases, contributing to improved model generalization (Section 5.2). This process is depicted in the third row of Figure 3 and the prompts for this are in Appendix A.8.

**Targeted Plan Augmentation:** While this large-scale data generation enhanced diversity and reduced underfitting for the PLANNER, it is not adaptive, and doesn't take into account what kinds of tasks are more difficult for the model in training. A key advantage of our approach is that we have explicit control over the data generation by allowing us to analyze the model failures and selectively refine the dataset.

Motivated by previous works (Lee et al., 2024; Qi et al., 2024; Yang et al., 2024b) around adaptive curriculum-learning, we ran our model through a held-out validation set which revealed several failure patterns in model performance. From there, our goal was to identify training data examples that seemed relevant to the failure patterns, instances where, if the model had seen more similar examples during training, it might improve performance on these tasks. To this end, we used an LLM to classify training data points that could be relevant to the failure nodes on each website and used them as seed data to generate 5,000 more synthetic plans. The prompts used to do this are in Appendix A.10 for classification and Appendix A.11 for generation. As we can see in Table 1, this targeted plan augmentation was able to significantly improve performance.

## 5. Results

### 5.1. Experimental Setup

- **Environment:** We run ablations on PLAN-AND-ACT using WebArena-Lite (Koh et al., 2024), a benchmark containing 165 test cases across diverse websites including OpenStreetMap, Reddit, GitLab, a content management system (CMS), and OneStopShop (OSS). WebArena-Lite uses a binary success metric (1 for complete task success, 0 for failure) and provides training data while being more computationally efficient than the full WebArena (Zhou

et al., 2023) benchmark. We also evaluate PLAN-AND-ACT on the full WebArena dataset as well as the Web-Voyager (He et al., 2024a) dataset, which is a dynamic, realworld web dataset. See Section A.2 and Section A.3.

- **Models:** For our primary PLAN-AND-ACT framework, we utilize LLaMA-3.3-70B-Instruct model by fine-tuning separate instances for both the PLANNER and EXECUTOR components. For our dynamic replanning experiments, we use a LLaMA-3.3-70B-Instruct model fine-tuned using LoRA (Hu et al., 2021) (due to computational constraints). Each component is trained on our synthesized datasets as described in previous sections. We use GPT-4o as the Synthetic Query Generator (Section 4.1), Plan Generator (Section 4.2) and Synthetic Plan Generator (Section 4.3). We use WebRL-Llama-3.1-70B (Qi et al., 2024) as the actor model and ORM-Llama-3.1-8B (Qi et al., 2024) as the filter model for filtering for successful trajectories. For generating CoT traces Section 3.4, we used DeepSeek-R1-Distill-Llama-70B (Guo et al., 2025) as the teacher model.

- **Baselines:** We compare PLAN-AND-ACT against several strong baselines to evaluate its effectiveness. These include zero-shot LLaMA-3.3-70B-Instruct without any fine-tuning, LLaMA-3.3-70B-Instruct fine-tuned specifically on the WebArena-Lite training set (ReAct-style prompting), and the WebRL-Llama-3.1-70B model, which is the current SOTA model on WebArena-lite. On Webarena-lite, we also compared against GPT-4-Turbo, GPT-4o, AWM (Wang et al., 2024b), and WebPilot (Zhang et al., 2024b). For the full WebArena dataset, we evaluated against NNetNav (Murty et al., 2024), AutoWebGLM (Lai et al., 2024), WebPilot, and AgentOccam (Yang et al., 2024a). For the WebVoyager dataset, we evaluated against NNetNav, OpenWebVoyager (He et al., 2024b), Wilbur(Lutz et al., 2024), and Agent-E (Abuelsaad et al., 2024). These models were evaluated with a success rate metric, which requires complete task completion for a positive score.

- **Hyperparameters:** The hyperparameters we used to train our PLANNER and EXECUTOR modules are found in Table 5. For the data generation in Section 4.1 and Section 4.3, we use 5 seed data points to generate 10 new synthetic data points.

### 5.2. Static PLANNER Results

Table 1 shows the results of our experiments.

The columns represent a different versions of the EXECUTOR (LLaMA-3.3-70B). The first column is a base EXECUTOR, which is not finetuned. The second column has an EXECUTOR that was trained only on 1,113 WebArena-lite

*Table 1.* Task success rate (SR) of PLAN-AND-ACT on WebArena-Lite, a human-verified subset of WebArena. The rows represent incremental improvements to the PLANNER, while the columns show results for different EXECUTOR. For the executor, the first column is a base prompted EXECUTOR, the second column is for an EXECUTOR finetuned on WebArena-lite data, and the third column shows results when finetuned on both WebArena-lite training data and the 923 synthetically generated data from Section 4.1. The first row shows the results without a PLANNER. For the results in this row, the Executors were trained ReAct style, with no plans. The second to sixth row shows reported results from WebRL (Qi et al., 2024) including the current SOTA on WebArena-lite. The seventh row shows the result when using a base zero-shot PLANNER. The eigth row adds finetuning with the WebArena-lite data and the ninth row adds finetuning with the additional data generated in Section 4.1. The tenth and eleventh row are after finetuning with the 10,000 synthetic plans generated in Section 4.3 and with the additional 5,000 synthetic plans generated by Targeted Augmentation. The 12th row shows the results after we introduce dynamic replanning into the architecture. The last row shows the results after adding CoT reasoning. Scores are averaged across all websites in the WebArena environment.

| PLANNER Design | EXECUTOR Design | | |
| --- | --- | --- | --- |
| | Base | + Finetuning | + Synthetic Traj. |
| No Planner | 9.85 | 36.36 | 36.97 |
| GPT-4-Turbo | - | - | 17.6* |
| GPT-4o | - | - | 13.9* |
| AWM + GPT-4-0613 (Wang et al., 2024b) | - | - | 35.5* |
| WebPilot + GPT-4o (Zhang et al., 2024b) | - | - | 37.2* |
| WebRL-3.1-70B (Qi et al., 2024) | - | - | 49.1* |
| Base | 14.21 | 17.16 | 23.63 |
| + Finetuning | 22.42 | 16.36 | 20.60 |
| + Synthetic Trajectories (Section 4.1) | 24.24 | 27.28 | 30.30 |
| + Plan Expansion (Section 4.3) | 27.10 | 38.18 | 39.40 |
| + Targeted Augmentation (Section 4.3) | 29.63 | 42.42 | 43.63 |
| + Dynamic Replanning (Section 3.3) | 44.24 | 48.48 | 53.94 |
| + CoT (PLAN-AND-ACT) (Section 3.4) | - | - | **57.58** |

training data points, and the third column being an EXECU-TOR trained on both the WebArena-lite training data as well as the 923 synthetically generated action trajectories from Section 4.1.

**NO PLANNER.** The first row shows the results for each of these Executors when trained with the baseline ReAct prompt (Yao et al., 2022b) without a PLANNER. As we can see, doubling the amount of action trajectory data does not significantly improve performance, showing a 0.61% improvement from just training on the WebArena-lite data. We cited this as motivation in Section 4.1 to focus on improving the PLANNER through plan generation.

**Base PLANNER.** The seventh row shows the results of enhancing each EXECUTOR with a base PLANNER (LLaMA-3.3-70B), which is not finetuned. As we can see, the performance improves for the base EXECUTOR, but fails to improve over the baseline for the trained Executors. What this can be attributed to is that since the PLANNER is not trained on data that is grounded to these specific websites, these plans are suboptimal and can confuse the EXECUTOR.

**Finetuned PLANNER.** The eighth row shows the results of using a PLANNER that has been finetuned only on the 1,113 plans in the WebArena-lite training data. As we can

see, naively finetuning the PLANNER did not improve performance for the finetuned EXECUTOR. Here, we found that the PLANNER was overfitting to the training data and did not generalize to general plans for new, unseen tasks.

**Finetuned PLANNER with data expansion.** The ninth through eleventh row show the results when iteratively different synthetic data augmentation strategies. The ninth row shows the results after augmenting the PLANNER with extra synthetic trajectories from Section 4.1. The tenth row shows the results after using these grounded trajectories to generate 10,000 synthetic query-plan pairs from Section 4.3, and the eleventh row shows the performance after also adding in the 5,000 targeted synthetic query-plan pairs.

From these rows, we can see that a properly trained PLAN-NER consistently improves performance across all of the different Executors. Even with a base EXECUTOR, adding a PLANNER increases success rate from 9.85% to 29.63%. This validates our core hypothesis that explicit planning helps bridge the gap between high-level user intentions and low-level actions.

The impact of generated data expansion is particularly notable. Each expansion of the training dataset yields performance improvements, with the most substantial gains

coming from the addition of the 10,000 directly generated plans, increasing success rates by approximately 10 percentage points. The failure-analysis-guided examples provided a final boost of 4-5 percentage points, highlighting the value of targeted data generation. Interestingly, the EXECUTOR's performance scales with training data size, but shows diminishing returns after the initial 1,113 examples, suggesting that the bottleneck may lie more in plan quality than action execution.

### 5.3. Dynamic Replanning Results

Despite these improvements, our detailed error analysis revealed fundamental limitations in the static PLANNER architecture. While the EXECUTOR performed well on concrete, deterministic tasks like navigating to specific pages, sorting tables, or posting comments on Reddit, it struggled with tasks requiring analysis of dynamic content. This aligns with our hypothesis in Section 3.3, that a static PLANNER would push some complex reasoning onto the EXECUTOR when it should focus solely on grounding abstract plans into specific actions. More explicit examples of this can be found in Appendix A.5.

#### 5.3.1. REPLANNING RESULTS

To address the previous limitation we finetuned the PLANNER with additional replanning data (as discussed in Section 3.3). The results are shown in the 12th row of Table 1. As we can see, the addition of this capability significantly increases the performance of the model by 10.31% over the static PLANNER, achieving 53.94% accuracy on WebArena-Lite. Notably, this result surpasses the previous SOTA WebRL-3.1-70B on the second row by 4.84%. You can see some explicit examples where the PLANNER with dynamic planning is able to refine and improve the plan in Section A.5, and a comprehensive breakdown of the performance by website can be found in Figure 4.

Importantly, even with a Base EXECUTOR (which has not been finetuned at all), we were able to significantly improve the performance of the model by 34.39%, achieving 44.24% accuracy, just by providing a high-quality and dynamic plan. This result highlights the importance of explicit planning and justifies our framework with separate PLANNER and EXECUTOR, demonstrating that a well-formed plan can substantially enhance performance even with an untrained EXECUTOR. WebArena and WebVoyager results can be found in Section A.2 and Section A.3.

## 6. Conclusion

It has been shown that separating high-level reasoning (PLANNER) from low-level execution (EXECUTOR), improves alignment between user queries and executable ac-

tions, enhancing task consistency and adaptability to dynamic environments. However, a major challenge is that out-of-the-box LLMs are not efficient at generating accurate plans for environments outside their pretraining distribution. In this work, we introduced PLAN-AND-ACT, a novel framework that enhances LLM agents' ability to tackle complex, long-horizon tasks through scalable synthetic data generation.

A key advantage of our method is its efficiency in data generation. Our pipeline generated 15,000 synthetic training examples in under an hour using GPT-4o, whereas environment interaction would take days or weeks. This scalability allowed us to match state-of-the-art performance with simple supervised fine-tuning.

Our results demonstrate that PLAN-AND-ACT significantly outperforms existing methods in web-based navigation tasks. Through synthetic data generation, plan expansion, and targeted refinement, our framework consistently improves success rates. Dynamic replanning further enhances model robustness by adapting strategies based on real-time observations.

By focusing on improving the planning component while keeping a standard EXECUTOR, we demonstrate our approach's potential. This modularity suggests future work could enhance performance by improving the EXECUTOR while maintaining our efficient planning framework. Beyond web navigation, our framework holds promise for broader applications in various digital environments requiring long-horizon decision making.

### Limitations

One main drawback is that Action Trajectory Generation Section 4.1 does depend on having a baseline model that can successfully complete the web tasks. The synthetic data generation pipeline introduced in Section 4.3 is able to mitigate some of these concerns with a sufficient amount of training data. However, for datasets that do not have any training data, such as WebVoyager, the pipeline will depend on having a base model to collect trajectories.

Furthermore, our current framework does dynamic replanning (3.3) after every action, which can be inefficient and slow down performance. Future work can address these concerns by having the EXECUTOR decide when it needs to replan, or by having the PLANNER delegate tasks to separate subagents.

### Acknowledgments

We thank Chris Joseph John and Anshul Verma for their help with the running of some of the benchmarks. We acknowledge gracious support from Apple team, as well as Nvidia for providing GPU hardware. We also appreciate the

support from Microsoft through their Accelerating Foundation Model Research. Furthermore, we appreciate support from Google Cloud, the Google TRC team, and specifically Jonathan Caton, Divvy Thakkar, and Prof. David Patterson. Prof. Keutzer's lab is sponsored by the Intel corporation, Intel One-API, Intel VLAB team, the Intel One-API center of excellence, as well as funding through BDD, BAIR, and Furiosa. We appreciate great feedback and support from Ellick Chan, Saurabh Tangri, Andres Rodriguez, and Kittur Ganesh. Sehoon Kim and Suhong Moon would like to acknowledge the support from the Korea Foundation for Advanced Studies (KFAS). Hiroki Furuta is supported by JSPS KAKENHI Grant Number JP22J21582. Our conclusions do not necessarily reflect the position or the policy of our sponsors, and no official endorsement should be inferred.

## Impact Statement

This work aims to improve the capabilities of large language model-based agents in performing complex, multi-step tasks by enhancing their planning abilities. Our framework could positively impact productivity and accessibility by enabling more effective AI assistants for web-based tasks. However, improved autonomous web agents could also raise privacy concerns and potentially automate certain jobs, particularly those involving routine online workflows. We believe these advances should be developed alongside appropriate safeguards and with consideration for economic transitions. Our synthetic data generation methods may also help reduce the data collection and annotation burden for training specialized agents, supporting more efficient AI development practices. Future research should explore the fairness and robustness of these planning systems across different user contexts and application domains.

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

# A. Appendix

## A.1. Chain of Thought Results

| Base Model | CoT | Performance (%) |
|---|---|---|
| Llama-3.3-70B | ✗ | 53.94 |
| Llama-3.1-8B | ✓ | 53.33 |
| QWQ-32B | ✓ | 54.88 |
| Llama-3.3-70B | ✓ | **57.58** |

*Table 2.* Comparison of PLAN-AND-ACT models with/without Chain-of-Thought. The first row shows the performance using a 70B model without the CoT data while the other rows show the performance using the CoT data.

In the 13th row of Table 1, it shows the result of adding CoT reasoning to the PLANNER and EXECUTOR. As we can see, it improves performance by 4.36% and sets a new SOTA of 57.58% on WebArena-lite.

In order to quantify the improvement from using CoT reasoning, we also finetuned a Llama-3.1-8B-instruct model and a QWQ-32B on the exact same data as the model in the 13th row in Table 2. As we can see, the 8B model on the second row performs on par with the non-CoT 70B model on the third row, which shows just how important CoT is.

## A.2. WebArena Results

| Method | Base Model | Acc. (%) |
|---|---|---|
| NNetNav (Murty et al., 2024) | Llama-3.1-8b | 16.3 |
| AutoWebGLM (Lai et al., 2024) | ChatGLM3-6B | 18.2 |
| WebPilot (Zhang et al., 2024b) | GPT-4o | 37.2 |
| AgentOccam (Yang et al., 2024a) | GPT-4-Turbo | 43.1 |
| AgentOccam-Judge (Yang et al., 2024a) | GPT-4-Turbo | 45.7 |
| PLAN-AND-ACT | Llama-70B | 45.7 |
| PLAN-AND-ACT | QWQ-32B | **48.15** |

*Table 3.* Comparison of methods on the WebArena benchmark. As you can see, PLAN-AND-ACT performs on-par with other prior work.

We also evaluated our approach on the full WebArena dataset in Table 3. As we can see, PLAN-AND-ACT performs better or on-par with most other prior work.

## A.3. WebVoyager Results

| Technique | Base Model | Acc. (%) |
|---|---|---|
| NNetNav (Murty et al., 2024) | Llama-3.1-8b | 34.2 |
| OpenWebVoyager (He et al., 2024b) | Idefics2-8b-inst. | 27.4 |
| WebVoyager (He et al., 2024a) (text) | GPT-4-Turbo | 44.3 |
| Wilbur (Lutz et al., 2024) | GPT-4-Turbo | 52.6 |
| WebVoyager (He et al., 2024a) | GPT-4-Turbo | 57.1 |
| PLAN-AND-ACT | Llama-3.1-8b | 58.08 |
| Agent-E (Abuelsaad et al., 2024) | GPT-4-Turbo | 73.1 |
| PLAN-AND-ACT | QWQ-32B | **81.36** |

*Table 4.* Comparison of techniques on the WebVoyager benchmark. PLAN-AND-ACT outperforms all open-source prior work and sets a new text-only SOTA on WebVoyager.

Furthermore, we evalauted PLAN-AND-ACT on the WebVoyager benchmark by finetuning a llama-3.1-8B model as well as a QWQ-32B model, which you can see in Table 4. Our goal was to evaluate our approach on real-world webtasks, as opposed to the simulator based tasks in WebArena.

Since WebVoyager does not have any training data, we used the text-only WebVoyager model with GPT-4o to collect 1500 synthetic trajectories (Section 4.1) and used QWQ-32B to annotate the plans (Section 4.2), CoT reasoning (Section 3.4) and to generate 10k synthetic plans (Section 4.3).

Our 8B model out performs all previous open source models and our 32B model out performs all prior work and sets a new SOTA for text-only WebVoyager.

### A.4. Planner and Executor Output Examples

- Task: "From my stay at Homewood Suites Southpointe, what's the estimated driving time to reach PPG Paints Arena?"

    – Planner Output

```
## Step 1
Reasoning: From the initial HTML state, I observe that I am on the
    OpenStreetMap website. The page includes a search input field and a
    direction sign icon that likely allows users to find directions
    between two points. The first step is to initiate the process of
    finding directions.
Step: Initiate the process of finding directions.

## Step 2
Reasoning: After initiating the direction-finding process, the next
    step is to input the starting location, which is 'Homewood Suites
    Southpointe'. This will set the starting point for the directions.
Step: Input 'Homewood Suites Southpointe' as the starting location.

## Step 3
Reasoning: With the starting location set, the next step is to input
    the destination, which is 'PPG Paints Arena'. This will set the
    endpoint for the directions.
Step: Input 'PPG Paints Arena' as the destination.

## Step 4
Reasoning: After setting both the starting location and the
    destination, the next step is to initiate the search for
    directions. This will generate the directions between the two
    points.
Step: Initiate the search for directions.

## Step 5
Reasoning: Once the directions are generated, the final step is to
    analyze the results to determine the estimated driving time from
    Homewood Suites Southpointe to PPG Paints Arena. The exact time
    will be displayed on the results page.
Step: Analyze the results to determine the estimated driving time from
    Homewood Suites Southpointe to PPG Paints Arena.
```

    – Executor Output
        * Output 1:

```
# Element: the direction sign on the right side of the Go icon
do(action="Click", element="15")}
```

        * Output 2:

```
# Element: The field at the beginning, located in the upper left
    corner
do(action="Type", argument="Homewood Suites Southpointe",
```

```
                        element="13")}
```

* Output 3:

```
# Element: The field labeled 'To' at the top left
do(action="Type", argument="PPG Paints Arena", element="14")}
```

* Output 4:

```
# Element: The Go button
do(action="Click", element="16")}
```

* Output 5:

```
# Note: Distance: 29km. Time: 0:34.
exit(message="The estimated driving time from Homewood Suites
    Southpointe to PPG Paints Arena is 0:34.")
```

- Task: "Create a shipping report from 08/05/2022 to 03/01/2023"

  - Planner Output

```
## Step 1
Reasoning: From the initial HTML state, I observe that I am on the
    Magento Admin Panel homepage. The page has a left sidebar
    navigation menu with several sections including 'Dashboard',
    'Sales', 'Catalog', 'Customers', 'Marketing', 'Content', 'Reports',
    'Stores', and 'System'. Each section appears to be clickable, and
    some have nested subsections visible through expandable lists. The
    main content area shows a dashboard with various widgets including
    revenue statistics, tax information, shipping details, and quantity
    metrics. To create a shipping report, we need to navigate to the
    'Reports' section in the left sidebar and then access the
    'Shipping' report option.
Step: Navigate to the 'Reports' section and access the 'Shipping'
    report option.

## Step 2
Reasoning: To generate the shipping report for the specified date
    range, we need to input the 'From' and 'To' dates. The 'From' date
    should be set to 08/05/2022 and the 'To' date should be set to
    03/01/2023. This will ensure that the report covers the desired
    period.
Step: Input the date range for the shipping report from 08/05/2022 to
    03/01/2023.

## Step 3
Reasoning: After setting the date range, we need to generate the report
    by clicking the 'Show Report' button. This will display the
    shipping report for the specified period.
Step: Generate the shipping report by clicking the 'Show Report' button.

## Step 4
Reasoning: Once the report is generated, we need to ensure that all
    relevant information is visible. This step ensures that the
    complete report details are accessible.
Step: Ensure that the complete shipping report details are visible.
```

```
## Step 5
Reasoning: Finally, we need to exit the task by providing a message
    that the shipping report for the specified date range has been
    created. This step concludes the task and informs the user of the
    successful completion.
Step: Exit with a message indicating that the shipping report for the
    specified date range has been created.
```

– Executor Output

* Output 1:

```
# Element: the 'Reports' tab in the left sidebar menu
do(action="Click", element="7")}
```

* Output 2:

```
# Element: the 'Shipping' link under the 'Sales' section
do(action="Click", element="18")}
```

* Output 3:

```
# Element: the 'From' input field in the 'Filter' section
do(action="Type", argument="08/05/2022", element="24")}
```

* Output 4:

```
# Element: the 'To' date input field in the 'Filter' section
do(action="Type", argument="03/01/2023", element="27")}
```

* Output 5:

```
# Element: the 'Show Report' button located at the top right area of
    the 'Shipping Report' section
do(action="Click", element="16")}
```

* Output 6:

```
do(action="Scroll Down")
```

* Output 7:

```
exit(message="The shipping report from 08/05/2022 to 03/01/2023 has
    been created.")
```

## A.5. Replanner Examples

### A.5.1. QUERY REFINEMENT

This section presents an example of how a replanner refines an initial plan when an execution failure occurs.

**Task:** "How long does it take to walk from Carnegie Museum of Art to a library at CMU?"

- Initial Plan

```
## Step 1
Reasoning: From the initial HTML state, I observe that I am on the
    OpenStreetMap website. The page includes a search input field. The
    first step is to search for libraries at CMU to identify which one
    we'll get directions to.
Step: Search for libraries at CMU.

## Step 2
Reasoning: After finding the libraries at CMU, we need to analyze the
    search results to identify the available libraries and select one for
    getting directions.
Step: Review the search results and select a library at CMU.

## Step 3
Reasoning: Now that we have identified a specific library, we can initiate
    the process of finding directions between the two points.
Step: Initiate the process of finding directions.

## Step 4
Reasoning: After initiating the direction-finding process, we need to input
    the starting location, which is 'Carnegie Museum of Art'.
Step: Input 'Carnegie Museum of Art' as the starting location.

## Step 5
Reasoning: With the starting location set, we need to input the selected
    library as the destination.
Step: Input the selected CMU library as the destination.

## Step 6
Reasoning: To ensure that the directions provided are for walking, we need
    to select the appropriate option for walking directions.
Step: Select the option for walking directions.

## Step 7
Reasoning: After all parameters are set, we can analyze the results to
    determine the walking time between the two locations.
Step: Analyze the results to determine the walking time from Carnegie
    Museum of Art to the selected CMU library.
```

- Executor's Actions: The executor attempts to search for libraries at CMU by performing the following actions:

```
# Element: the search bar
do(action="Search", argument="Library at CMU", element="13")

# Element: the search button
do(action="Click", element="13")
```

However, upon executing this step, the system receives the following **HTML response**, which indicates that no search results were found:

```
<html>
    ...
    <li id="17" data-text="No results found"> No results found </li>
    ...
</html>
```

This failure necessitates a refinement of the plan.

• Replanning - Refining the Search Query: Since the initial search query ("Library at CMU") returned no results, the replanner modified the query to a more general term: **"Library near CMU"** to improve the likelihood of retrieving relevant results.

   – Refined Plan

```
## Step 1
Reasoning: From the current HTML state, I observe that the page shows
    search results with 'No results found' (element with id='17'). The
    page contains a search bar (id='12') and a Go button (id='13') for
    initiating new searches. There's also a direction sign icon
    (id='14') that likely allows for finding directions between two
    points. The main content area provides information about
    OpenStreetMap and its contributors. Looking at the previous
    actions, I can see that the user attempted to search for 'Library
    at CMU' and clicked the Go button, but this broad search query
    including 'Library at CMU' returned no results. This indicates that
    we need to modify our approach by simplifying the search terms.
    Given these observations and the failed previous search attempt, we
    need to refine our search query by removing the 'Library at CMU'
    specification and focusing more specifically on libraries near CMU
    to get more relevant results.
Step: Refine the search query by focusing on libraries near CMU.

## Step 2
Reasoning: After refining the search query, we need to analyze the
    search results to identify potential libraries that are near CMU.
    This will help in determining the closest library based on the
    search results.
Step: Analyze the search results to identify potential libraries near
    CMU.

## Step 3
Reasoning: With the list of potential libraries identified, the next
    step is to get directions to each library from Carnegie Museum of
    Art. This will help in determining the distance and time required
    to reach each library, allowing us to identify the closest one.
Step: Get walking directions to each identified library from Carnegie
    Museum of Art.

## Step 4
Reasoning: After obtaining the distances and times for each library,
    the final step is to identify the closest library based on the
    recorded distances and provide the result to the user.
Step: Identify the closest library based on the recorded distances and
    provide the result to the user.
```

   – Executor's Next Action According to the Refined Plan: To reflect the updated strategy, the executor modifies its search action:

```
do(action="Search", argument="Library near CMU", element="12")
```

A.5.2. ANALYZING SEARCH RESULTS AND MEMORY (AGGREGATING RESULTS ACROSS MULTIPLE PAGES)

This section presents an example of how a replanner refines an initial plan when a broad high-level step, such as "analyze the search results," needs to be refined into concrete actions.

**Task:** "Present the monthly count of successful orders from Jan to December 2022 in MM:COUNT format"

- Initial Plan

```
## Step 1
Reasoning: From the initial HTML state, I observe that I am on the Magento
    Admin Panel homepage. The page has a left sidebar navigation menu with
    several sections including 'Dashboard', 'Sales', 'Catalog',
    'Customers', 'Marketing', 'Content', 'Reports', 'Stores', and 'System'.
    Each section appears to be clickable and some have nested subsections
    visible through expandable lists. The main content area shows a
    dashboard with various widgets including revenue statistics, tax
    information, shipping details, and quantity metrics. To access the
    order reports, we need to navigate through the 'Reports' section in the
    left sidebar, where we can then find and select the 'Orders' report
    option under the 'Sales' section.
Step: Navigate to the 'Orders' report section under the 'Reports' section.

## Step 2
Reasoning: To generate the report for the specified period, we need to set
    the date range from January 1, 2022, to December 31, 2022. This
    involves entering the start and end dates in the respective date picker
    input fields.
Step: Set the date range from January 1, 2022, to December 31, 2022.

## Step 3
Reasoning: To ensure the report is generated for the correct period, we
    need to select the 'Month' option from the 'Period' dropdown. This will
    aggregate the data on a monthly basis.
Step: Select the 'Month' option from the 'Period' dropdown.

## Step 4
Reasoning: After setting the date range and period, we need to generate the
    report by clicking the 'Show Report' button. This will display the
    order report for the specified period.
Step: Generate the report by clicking the 'Show Report' button.

## Step 5
Reasoning: Once the report is generated, we need to analyze the results to
    identify the monthly count of successful orders from January to
    December 2022. This involves viewing the report and noting the counts
    for each month.
Step: Analyze the report results to identify the monthly count of
    successful orders from January to December 2022.

## Step 6
Reasoning: After identifying the monthly counts, we need to provide the
    results in the 'MM:COUNT' format. This involves exiting with a message
    containing the monthly counts for each month from January to December
    2022.
Step: Exit with a message containing the monthly counts of successful
    orders from January to December 2022 in the 'MM:COUNT' format.
```

- Executor's Actions: The executor follows the plan up until the broad "analyze the report results" step:

```
# Element: the 'REPORTS' sectiondo(action="Click", element="7")
```

```
# Element: the 'Orders' report optiondo(action="Click", element="15")

# Element the 'From' datedo(action="Type", argument="01/01/2022",
    element="24")

# Element the 'To' datedo(action="Type", argument="12/31/2022",
    element="27")

# Element: 'Month' in the dropdown
do(action="Select Dropdown Option", argument="Month", element="22")

# Element: 'Show Report'
do(action="Click", element="16")
```

The executor reaches the analysis step, but the broad instruction to "analyze the report" needs refinement.

- Replanning - Refining the Analysis Task: The replanner observes that the executor requires step-by-step guidance to extract the relevant order counts. It refines the broad analysis step into structured data extraction.

- First Replan
  - First step of the replan:

    ```
    ## Step 1
    Reasoning: The report results are now visible, and I need to analyze
        the order counts for each month. The report is structured as a
        list, where each row contains a month and the corresponding order
        count. The first step is to start extracting data from the
        initially visible rows. The extracted counts so far are:
    01:11, 02:16, 03:14, 04:8
    Step: Extract the order counts from the initially visible rows.
    ```

  - Next executor action:

    ```
    do(action="Scroll Down")
    ```

- Second Replan
  - First step of the replan:

    ```
    ## Step 1
    Reasoning: After capturing the first set of order counts, I need to
        scroll down to reveal more entries. As new rows become visible, I
        will extract and record their data. The additional extracted counts
        are:
    05:8, 06:13, 07:9, 08:8, 09:10, 10:4
    Step: Scroll down and extract the next set of order counts.
    ```

  - Next executor action:

    ```
    do(action="Scroll Down")
    ```

- Third Replan
  - First step of the replan:

```
## Step 1
Reasoning: The final set of order counts needs to be retrieved. I will
    scroll down again, ensuring I capture all remaining visible data.
    The final extracted counts are:
11:11, 12:10
Step: Perform a final scroll to capture any remaining order counts.
```

– Next executor action:

```
exit(message="The monthly count of successful orders from Jan to
    December 2022 is: 01:11, 02:16, 03:14, 04:8, 05:8, 06:13, 07:9,
    08:8, 09:10, 10:4, 11:5, 12:10")
```

## A.6. Planner Prompt

Planner has a system prompt where its high-level goals are listed; in the user message, it sees the current user query and the initial HTML state.

### A.6.1. PLANNER SYSTEM PROMPT

```
## Goal
You are the Global Planner agent, an expert plan generator for web navigation
    tasks. You will be proivded with the following information:
- **User Query**: The web task that you are required to generate a global plan
    for.
- **Initial HTML State**: The initial HTML state of the web page.

You are responsible for analyzing the usery query and the initial HTML state
    to generate a structured, step-by-step global plan that outlines the
    high-level steps to complete the user query. The global plan that you
    generate shouldn't directly describe low-level web actions such as clicks
    or types (unless necessary for clarity) but outline the high-level steps
    that encapsulate one or more actions in the action trajectory, meaning each
    step in your plan will potentially require multiple actions to be
    completed. Your global plan will then be handed to an Executor agent which
    will perform low-level web actions on the webpage (click, type, hover, and
    more) to convert your global plan into a sequence of actions and complete
    the user query.

## Expected Output Format
The global plan you generate should be structured in a numbered list format,
    starting with '## Step 1' and incrementing the step number for each
    subsequent step. Each step in the plan should be in this exact format:
```
## Step N
Reasoning: [Your reasoning here]
Step: [Your step here]
```

Here is a breakdown of the components you need to include in each step of your
    global plan as well as their specific instructions:
- **Reasoning**: In this section, you should explain your reasoning and
    thought process behind the step you are proposing. It should provide a
    high-level justification for why the actions in this step are grouped
    together and how they contribute to achieving the overall goal. Your
    reasoning should be based on the information available in the user query
    (and potentially on the initial HTML state) and should guide the Executor
    agent in understanding the strategic decision-making process behind your
    global plan.
```

- **Step**: In this section, you should provide a concise description of the
    global step being undertaken. Your step should summarize one or more
    actions as a logical unit. It should be as specific and concentrated as
    possible. Your step should focus on the logical progression of the task
    instead of the actual low-level interactions, such as clicks or types.

## Guidelines:
- Ensure every action and reasoning aligns with the user query, the webpage at
    hand, and the global plan, maintaining the strict order of actions.
- Minimize the number of steps by clustering related actions into high-level,
    logical units. Each step should drive task completion and avoid unnecessary
    granularity or redundancy. Focus on logical progression instead of
    detailing low-level interactions, such as clicks or UI-specific elements.
- Provide clear, specific instructions for each step, ensuring the executor
    has all the information needed without relying on assumed knowledge. For
    example, explicitly state, 'Input 'New York' as the arrival city for the
    flights,' instead of vague phrases like 'Input the arrival city.'
- You can potentially output steps that include conditional statements in
    natural language, such as 'If the search results exceed 100, refine the
    filters to narrow down the options.' However, avoid overly complex or
    ambiguous instructions that could lead to misinterpretation.

## High-level Goals Guidelines:
- Focus on high-level goals rather than fine-grained web actions, while
    maintaining specificity about what needs to be accomplished. Each step
    should represent a meaningful unit of work that may encompass multiple
    low-level actions (clicks, types, etc.) that serve a common purpose, but
    should still be precise about the intended outcome. For example, instead of
    having separate steps for clicking a search box, typing a query, and
    clicking search, combine these into a single high-level but specific step
    like "Search for X product in the search box".
- Group related actions together that achieve a common sub-goal. Multiple
    actions that logically belong together should be combined into a single
    step. For example, multiple filter-related actions can be grouped into a
    single step like "Apply price range filters between $100-$200 and select
    5-star rating". The key is to identify actions that work together to
    accomplish a specific objective while being explicit about the criteria and
    parameters involved.
- Focus on describing WHAT needs to be accomplished rather than HOW it will be
    implemented. Your steps should clearly specify the intended outcome without
    getting into the mechanics of UI interactions. The executor agent will
    handle translating these high-level but precise steps into the necessary
    sequence of granular web actions.

## Initial HTML State Guidelines:
- Use the initial HTML of the webpage as a reference to provide context for
    your plan. Since this is just the initial HTML, possibly only a few of the
    initial actions are going to be taken on this state and the subsequent ones
    are going to be taken on later states of the webpage; however, this initial
    HTML should help you ground the plan you are going to generate (both the
    reasoning behind individual steps and the overall plan) in the context of
    the webpage at hand. This initial HTML should also help you ground the task
    description and the trajectory of actions in the context of the webpage,
    making it easier to understand the task.
- You MUST provide an observation of the initial HTML state in your reasoning
    for the first step of your global plan, including the elements, their
    properties, and their possible interactions. Your observation should be
    detailed and provide a clear understanding of the current state of the HTML
    page.

## Formatting Guidelines:
- Start your response with the '## Step 1' header and follow the format
    provided in the examples.

```
  - Ensure that each step is clearly separated and labeled with the '## Step N'
      header, where N is the step number.
  - Include the 'Reasoning' and 'Step' sections in each step.
```

### A.6.2. PLANNER USER MESSAGE

```
## User Query
{user_query}

## Initial HTML State
{initial_html_state}

You MUST start with the '## Step 1' header and follow the format provided in
    the examples.
```

## A.7. Executor Prompt

Executor follows the WebArena-Lite defined executor prompt where each user-assistant message pair represents an HTML-action round. The only addition we have is to the system prompt which describes what a plan is.

### A.7.1. EXECUTOR SYSTEM PROMPT

```
# Goal
You are the Executor Agent, a powerful assistant can complete complex web
    navigation tasks by issuing web actions such as clicking, typing,
    selecting, and more. You will be provided with the following information:
- **Task Instruction**: The web task that you are required to complete.
- **Global Plan**: A high-level plan that guides you to complete the web tasks.
- **Previous action trajectory**: A sequence of previous actions that you have
    taken in the past rounds.
- **Current HTML**: The current HTML of the web page.

Your goal is to use the Global Plan, the previous action trajectory, and the
    current observation to output the next immediate action to take in order to
    progress toward completing the given task.

# Task Instruction: {intent}

# Global Plan
The Global Plan is a structured, step-by-step plan that provides you with a
    roadmap to complete the web task. Each step in the Global Plan (denoted as
    '## Step X' where X is the step number) contains a reasoning and a
    high-level action that you need to take. Since this Global Plan
    encapsulates the entire task flow, you should identify where you are in the
    plan by referring to the previous action trajectory and the current
    observation, and then decide on the next action to take. Here is the Global
    Plan for the your task:

{global_plan}
```

## A.8. Plan Data Annotator Prompt

Similar to the planner prompt, there is a system prompt that defines the goals of the plan annotator; and the user message provides the user query, the initial HTML state, and the action trajectory for which the plan annotator needs to generate a plan.

A.8.1. PLAN DATA ANNOTATOR SYSTEM PROMPT

```
## Goal
You are the Global Planner agent, an expert plan generator for web navigation
    tasks. You will be proivded with the following information:
- **User Query**: The web task that you are required to generate a global plan
    for.
- **Initial HTML State**: The initial HTML state of the web page.
- **Trajectory**: A sequence of actions that represent a trajectory of a web
    navigation task. It is formatted as series of actions where each action
    first has a comment ('#') that describes the element to be interacted with
    or a note what provides some context about the action and the current task
    state. The action is then described with the do function, which takes two
    arguments: the action to be performed, the element to be interacted with,
    and sometimes an argument. The actions are numbered sequentially to
    indicate the order in which they should be executed.

You are responsible for analyzing initial HTML state and the trajectory
    provided below and producing a structured, step-by-step global plan that
    clusters multiple actions into the fewest number of logical steps possible.
    The global plan that you generate shouldn't describe fine-grained web
    interactions such as clicks or types but outline the high-level steps that
    encapsulate one or more actions in the trajectory, meaning each step in
    your plan will potentially require multiple actions to be completed. You
    will also be tasked to classify each action in the trajectory with one of
    the steps in your global plan. Each of your steps will be handed to another
    executor agent that will convert your step into fine-grained web
    interactions; hence, your steps should include every specific information
    needed for completing the task without assuming the executor agent has
    access to the whole task or trajectory.

## Expected Output Format
The global plan you generate should be structured in a numbered list format,
    starting with '## Step 1' and incrementing the step number for each
    subsequent step. Each step in the plan should be in this exact format:
```
## Step N
Reasoning: [Your reasoning here]
Description: [Description of the actions this step covers]
Step: [Your step here]
Actions: [list of action indexes associated with this step]
```

Here is a breakdown of the components you need to include in each step of your
    global plan as well as their specific instructions:
- **Reasoning**: In this section, you should explain your reasoning and
    thought process behind the step you are proposing. It should provide a
    high-level justification for why the actions in this step are grouped
    thogether and how they contribute to achieving the overall goal. Your
    reasoning should be based on the information available in the trajectory
    (and potentially on the initial HTML state) and should guide the executor
    agent in understanding the strategic decision-making process behind your
    global plan.

- **Description**: This section should include a brief description of the
    actions that are grouped together in this step. You should exactly copy the
    action descriptions from the trajectory without any modifications or
    additional information. This is to ensure that the executor agent can
    accurately map the actions to the global plan steps. Specifically, every
    action that you include in your description should include any '# Element',
    '# Note', or '# Exit' comments that are present in the trajectory as well
    as their corresponding 'do' functions.
```

- **Step**: In this section, you should provide a concise description of the
  global step being undertaken. Your step should summarize one or more
  actions from the trajectory as a logical unit. It should be as specific and
  concentrated as possible, without referring to any HTML or UI elements.
  Your step should focus on the logical progression of the task instead of
  the actual fine-grained interactions, such as clicks or types.

- **Actions**: This section should list the indexes of the actions associated
  with this step. One or more actions should be grouped under one broader
  logical step. The indices in this section should exactly match the indices
  of the actions in the trajectory.

## Examples
Here are some examples of the expected output format for the global plan where
    the input is the task description and the trajectory of actions taken to
    complete the task and the output is the structured global plan that
    clusters multiple actions into the fewest number of logical steps possible
    without sacrificing specificity:

{in_context_examples}

## Planning Guidelines:
- Ensure every action and thought aligns with the trajectory and global plan,
    maintaining the strict order of actions. Actions should be sequential, with
    no skipping or misalignment (e.g., avoid assigning non-consecutive actions
    like Step 1: [0,3,4], Step 2: [1,2]). Deviation from the trajectory's order
    will be PENALIZED!
- Minimize the number of steps by clustering related actions into high-level,
    logical units. Each step should drive task completion and avoid unnecessary
    granularity or redundancy. Focus on logical progression instead of
    detailing fine-grained interactions, such as clicks or UI-specific elements.
- Provide clear, specific instructions for each step, ensuring the executor
    has all the information needed without relying on assumed knowledge. For
    example, explicitly state, 'Input 'New York' as the arrival city for the
    flights,' instead of vague phrases like 'Input the arrival city.'
- You can potentially output steps that include conditional statements in
    natural language, such as 'If the search results exceed 100, refine the
    filters to narrow down the options.' However, avoid overly complex or
    ambiguous instructions that could lead to misinterpretation.

## High-level Goals Guidelines:
- Focus on high-level goals rather than fine-grained web actions, while
    maintaining specificity about what needs to be accomplished. Each step
    should represent a meaningful unit of work that may encompass multiple
    low-level actions (clicks, types, etc.) that serve a common purpose, but
    should still be precise about the intended outcome. For example, instead of
    having separate steps for clicking a search box, typing a query, and
    clicking search, combine these into a single high-level but specific step
    like "Search for X product".
- Group related actions together that achieve a common sub-goal. Multiple
    actions that logically belong together should be combined into a single
    step. For example, multiple filter-related actions can be grouped into a
    single step like "Apply price range filters between $100-$200 and select
    5-star rating". The key is to identify actions that work together to
    accomplish a specific objective while being explicit about the criteria and
    parameters involved.
- Focus on describing WHAT needs to be accomplished rather than HOW it will be
    implemented. Your steps should clearly specify the intended outcome without
    getting into the mechanics of UI interactions. The executor agent will
    handle translating these high-level but precise steps into the necessary
    sequence of granular web actions.
- Provide clear, specific instructions for each step, ensuring the executor
    has all the information needed without relying on assumed knowledge. For
    example, explicitly state, 'Input 'New York' as the arrival city for the

```
          flights,' instead of vague phrases like 'Input the arrival city.'
   - The action trajectory might include several "scroll down" actions necessary
       to locate or find an element, but you should not explicitly say "scroll
       down to find X" in your step description. Instead, you can use phrases like
       "locate X", "find Y", "look for Z", or similar phrases to represent the
       scroll actions in your step description. The act of scrolling is not part
       of the high-level goal but just implementation details, so you should not
       explicitly mention it in your step description.
   - Example:
     BAD plan (mentions scrolling):
     ```
     Step 1: Scroll down to find the 'Contact Us' button and click it
     Step 2: Scroll through the list to find the order numbered ID12345
     ```

     GOOD plan (avoids mentioning scrolling):
     ```
     Step 1: Locate the 'Contact Us' button and click it
     Step 2: Find the order numbered ID12345
     ```

   ## Initial HTML State Guidelines:
   - Use the initial HTML of the webpage as a reference to provide context for
       your plan. Since this is just the initial HTML, possibly only a few of the
       initial actions are going to be taken on this state and the subsequent ones
       are going to be taken on later states of the webpage; however, this initial
       HTML should help you ground the plan you are going to generate (both the
       reasoning behind individual steps and the overall plan) in the context of
       the webpage at hand. This initial HTML should also help you ground the task
       description and the trajectory of actions in the context of the webpage,
       making it easier to understand the task.
   - You MUST provide an observation of the initial HTML state in your reasoning
       for the first step of your global plan, including the elements, their
       properties, and their possible interactions. Your observation should be
       detailed and provide a clear understanding of the current state of the HTML
       page. Please refer to the examples for more information on how to do this.

   ## Formatting Guidelines:
   - Start your response with the '## Step 1' header and follow the format
       provided in the examples.
   - Ensure that each step is clearly separated and labeled with the '## Step N'
       header, where N is the step number.
   - Include the 'Reasoning', 'Actions that this step covers', 'Indices of
       actions', and 'Step' sections in each step.
```

A.8.2. PLAN DATA ANNOTATOR USER MESSAGE

```
## User Query
{goal_description}

## Initial HTML State
{initial_html_state}

## Trajectory
The following is a sequence of actions that represent a trajectory of a web
    navigation task. It is formatted as series of actions where each action
    first has a comment ('#') that describes the element to be interacted with
    or a note what provides some context about the action and the current task
    state. The action is then described with the do function, which takes two
    arguments: the action to be performed, the element to be interacted with,
```

```
        and sometimes an argument. The actions are numbered sequentially to
        indicate the order in which they should be executed:

{trajectory}
```

## A.9. Synthetic Plan Generator Prompt

Similarly, the synthetic plan generator also has a system prompt that presents the goals of the synthetic plan generator and also provides the seed data examples. The user message specifies how many synthetic plans to generate (from the seed data examples in the system prompt).

### A.9.1. SYNTHETIC PLAN GENERATOR SYSTEM PROMPT

```
# Goal
You are a Plan Data Generator that can generate new synthetic data to train a
    planner language model to be excellent at plan generation for web
    navigation tasks. The data that this model is going to trained (and hence
    the data you generate) is going to be in the following format:
- **Input**: A user query for a web navigation task.
- **Output**: A high-level global plan to accomplish the task.

 You will be given some examples on how the input-output pairs look like and
     your goal is to generate new data pairs that are similar to the examples
     given. Your goal is to increase the data diversity by covering a wide
     range of possible user queries while also grounding your data generation
     process on the specific website that the examples are based on. You
     shouldn't just copy the examples since that would not help the model
     generalize better but you also shouldn't generate data that is not
     possible on the website. You must use the given examples to infer what is
     possible on the website and ground your generated data on it.

# Expected Output Format
The input-output pairs you generate should be structured as follows:
```
## Data Pair {{i}}
User Query:
<user query>

Initial HTML State:
<index of the example whose initial HTML state you are starting from>

Global Plan:
<global plan>
```
where:
- `{{i}}` is the data pair number.
- `<user query>` is a brief description of the task that the user wants to
    accomplish on a website.
- `<index of the example whose initial HTML state you are starting from>` is
    the index of the example whose initial HTML state you are starting from.
    This is just an integer like 1, 3, etc.
- `<global plan>` is a high-level global plan that outlines the steps needed
    to accomplish the task.

# Instructions
Here are the guidelines to follow when generating the data:

## User Query Instructions
The User Query is a brief description of the task that the user wants to
    accomplish on a website. It should be concise and focused on the main goal
    of the task. The user query should provide enough context for an agent to
```

```
                   generate a high-level global plan to accomplish the task.

## Initial HTML State Instructions
- The Initial HTML State is the HTML representation of the webpage at the
    beginning of the task. It provides the context for the user query and the
    global plan. When generating new data, you should choose the initial HTML
    state of one of the examples that you want to start from and provide the
    index of the example whose initial HTML state you are starting from. This
    will ensure that the generated data is grounded in the context of the
    specific website and HTMLs that the examples are based on. You should only
    provide the index of the example whose initial HTML state you are starting
    from. For example, if you are starting from the second example's initial
    HTML state ('# Example 2'), you should provide '2' as the initial HTML
    state.
- When generating multiple data pairs, you should aim to use different
    examples' initial HTML states in a balanced way. While you don't need to
    use each HTML state exactly equally, you should ensure good coverage across
    all examples. Some HTML states may enable a wider range of user queries and
    can be used more frequently, but you shouldn't completely ignore or heavily
    underutilize any of the examples. The goal is to leverage the full range of
    possible HTML states and website functionalities shown in the examples.
- Aftering picking which HTML to start from, you MUST provide an observation
    of the initial HTML state in your reasoning for the first step of your
    global plan, including the elements, their properties, and their possible
    interactions. Your observation should be detailed and provide a clear
    understanding of the current state of the HTML page. Please refer to the
    examples for more information on how to do this.

## Global Plan Instructions
The Global Plan is a structured, step-by-step plan that provides a high-level
    overview of the actions that need to be taken to accomplish a web
    navigation task. The plan should be detailed enough to guide the user
    through the task but not too detailed that it becomes a step-by-step
    instruction. In other words, the global plan that you generate shouldn't
    directly describe low-level web actions such as clicks or types (unless
    necessary for clarity) but outline the high-level steps that encapsulate
    one or more actions in the action trajectory, meaning each step in your
    plan will potentially require multiple actions to be completed. Your global
    plan will then be handed to an Executor agent which will perform low-level
    web actions on the webpage (click, type, hover, and more) to convert your
    global plan into a sequence of actions and complete the user query.

### Global Plan Expected Output Format
The global plan you generate should be structured in a numbered list format,
    starting with '## Step 1' and incrementing the step number for each
    subsequent step. Each step in the plan should be in this exact format:
```
## Step N
Reasoning: [Your reasoning here]
Step: [Your step here]
```

Here is a breakdown of the components you need to include in each step of your
    global plan as well as their specific instructions:
- **Reasoning**: In this section, you should explain your reasoning and
    thought process behind the step you are proposing. It should provide a
    high-level justification for why the actions in this step are grouped
    together and how they contribute to achieving the overall goal. Your
    reasoning should be based on the information available in the user query
    (and potentially on the initial HTML state) and should guide the Executor
    agent in understanding the strategic decision-making process behind your
    global plan.

- **Step**: In this section, you should provide a concise description of the
    global step being undertaken. Your step should summarize one or more
```

```
      actions as a logical unit. It should be as specific and concentrated as
      possible. Your step should focus on the logical progression of the task
      instead of the actual low-level interactions, such as clicks or types.

## High-level Goals Guidelines:
- Focus on high-level goals rather than fine-grained web actions, while
    maintaining specificity about what needs to be accomplished. Each step
    should represent a meaningful unit of work that may encompass multiple
    low-level actions (clicks, types, etc.) that serve a common purpose, but
    should still be precise about the intended outcome. For example, instead of
    having separate steps for clicking a search box, typing a query, and
    clicking search, combine these into a single high-level but specific step
    like "Search for X product".
- Group related actions together that achieve a common sub-goal. Multiple
    actions that logically belong together should be combined into a single
    step. For example, multiple filter-related actions can be grouped into a
    single step like "Apply price range filters between $100-$200 and select
    5-star rating". The key is to identify actions that work together to
    accomplish a specific objective while being explicit about the criteria and
    parameters involved.
- Focus on describing WHAT needs to be accomplished rather than HOW it will be
    implemented. Your steps should clearly specify the intended outcome without
    getting into the mechanics of UI interactions. Another executor agent will
    handle translating these high-level but precise steps into the necessary
    sequence of granular web actions.

# Examples
Here are some examples you must utilize to understand what is possible on the
    website, what kind of actions are executable, what HTML elements are
    present on the website, and what kind of tasks you can generate data for.
    Remember:
1. You are required to take inspiration from these example but not exactly
    copy them since we want enough diversity to be able to cover a wide variety
    of use cases.
2. You shouldn't hallucinate or create non-existing elements or actions that
    are not possible on the website. If you make up something that is not
    possible on the website, you will be penalized. Your data needs to be
    grounded on the website and the examples given.

{examples_str}
```

## A.9.2. SYNTHETIC PLAN GENERATOR USER MESSAGE

```
Use the given examples to generate {how_many_to_generate_at_once} new data
    pairs. The data pairs you generate SHOULDN'T be similar to each other. They
    should be diverse and cover a wide range of possible user queries and tasks.

# Output Formatting
You should output the data pairs you generate in the following format:
```
## Data Pair {i}
User Query:
<user query>

Initial HTML State:
<index of the example whose initial HTML state you are starting from. Remember
    this is just an integer like 1, 3 etc.>

Global Plan:
<global plan>
```
```

```
# Remember
- You shouldn't hallucinate or create non-existing elements or actions that
    are not possible on the website. If you make up something that is not
    possible on the website, you will be penalized. Your data needs to be
    grounded on the website and the examples given.
- You are required to take inspiration from these examples but not exactly
    copy them since we want enough diversity to be able to cover a wide variety
    of use cases. However, while trying to create diverse data, you MUST avoid
    making up non-existing elements or actions that are not possible on the
    website.
- You MUST provide a detailed initial HTML state observation for the first
    step of your global plan.
```

## A.10. Training Data Failure Classification Prompt

For classifying the training data based on the failure classes we identified, we have the main system prompt which defines the goal of the classification model. Each website has its own failure classes. If the model classifies a training data point into any one of these classes, we keep that data point for the next round of synthetic data generation.

### A.10.1. MAIN SYSTEM PROMPT

```
## Goal
You are an expert classifier model tasked with classifying data points that
    were used to train a "Planner" model. This model was trained to take in a
    user query (or a task) related to common websites such as shopping
    websites, Reddit, GitLab, etc., and output a high-level global plan for
    completing that task. After training, we conducted a failure analysis to
    identify the types of errors the planner was most prone to.

Now, using the identified failure classes, we aim to label the training points
    of the global planner. The purpose of this classification is to determine
    which data points can be leveraged to generate synthetic data. This
    synthetic data will be used to retrain the planner, helping it correct its
    mistakes and avoid previous failures.

For each data point, you will receive:
- The website name: e.g., "shopping_admin"
- A user query (task): The user query or task that the planner is supposed to
    complete
- A ground truth global plan: The global planner was trained to generate this
    plan for the given user query.

Remember: The data points that will be given to you are going to be perfect
    (they are from the training data): They are going to be the best possible
    plans that the planner can generate. Hence, your job is not to classify the
    data point itself into a failure class but rather identify whether this
    data point is a good example to train the planner to generate better plans
    and which failure class it will potentially help the planner avoid.

Your job:
1) Read the given user query and the plan carefully
2) Identify what this data points is trying to do and what can the planner
    model learn from being trained on this data point and data points like it
3) Provide clear reasoning for your classification decision
4) Classify the data point into one of the known failure classes for that
    website or "Other" if no class fits; specifically, you should classify the
    failure class that this data point will help the planner avoid if it was
    trained on this data point and data points like it

Below is the set of possible classes for the website: {website.value}.
{classification_section_for_website}
```

```
General guidelines:
1. Carefully check the user query and plan
2. Match them against the class definitions
3. If none of the classes apply, label as "Other"
4. Provide your output in the following format:

## Reasoning
[Explain your thought process and why this example fits the chosen class]

## Classification
[Class label: "Class A", "Class B", "Other", etc.]

Please ensure your output follows this exact format.
```

Here are the prompts for the failure classification model for each website separately. Each failure class was identified by looking at our model's performance on the validation set:

A.10.2. SHOPPING ADMIN (CMS) FAILURE CLASSES

```
Here are the classes for the shopping_admin website:

# Shopping Admin Website Classes

## Class A: Search Query Optimization Failures

### Description
The planner fails to implement proper search query strategies, particularly:
- Using overly specific search terms without fallback to broader terms
- Not utilizing the search functionality effectively when simpler queries
    would work
- Missing critical search parameters or using irrelevant ones

### Training Data Needed
- Examples showing fallback to broader search terms when specific searches fail
- Cases demonstrating effective use of search functionality with simpler
    queries

### Example Tasks
1. "Show me the name of the customers who have expressed dissatisfaction with
    Chloe tank"
    - Error: Planner used exact "chloe tank" search instead of broader "chloe"
      search that would have found "chloe plastic tank"

2. "List the top 3 search terms in my store"
    - Error: Planner incorrectly included date filtering steps which don't
      exist in search terms report
    - Solution: Training data showing correct navigation of "search terms"
      report without date filtering

## Class B: Product Attribute Update Confusion

### Description
The planner confuses high-level status changes with specific attribute updates:
- Using "Change status" action instead of updating specific product attributes
- Attempting to modify stock/price/sale status through wrong interface elements

### Training Data Needed
- Examples showing correct attribute updates for sales status
- Cases demonstrating proper stock level modifications
- Examples distinguishing between status changes and attribute updates
```

```
### Example Tasks
1. "Mark all Hollister shirts on sale"
   - Error: Planner used general status change instead of specific sale
     attribute update
   - Solution: Training data showing how to update sale attributes
     specifically using 'update attributes' option

2. "Make all Aeno capri as out of stock"
   - Error: Planner tried using Enable/Disable status instead of stock
     attribute
   - Solution: More examples of updating product attributes vs changing status

## Class C: Review Analysis Navigation Failures

### Description
The planner fails to properly navigate and analyze product reviews:
- Missing steps to access product review sections
- Failing to specify review content examination steps
- Not including steps to gather specific review details

### Training Data Needed
- Examples showing navigation to product review sections
- Cases demonstrating proper review content analysis
- Examples of gathering specific review details

### Example Tasks
1. "Tell me the reasons why customers like Circe's products"
   - Error: Planner didn't include steps to access and analyze review content
   - Solution: Training data showing how to navigate to and analyze review
     sections

## Other
Description: If none of the above classes match.
```

### A.10.3. REDDIT FAILURE CLASSES PROMPT

```
# Reddit Website Classes

## Class A: Content Reposting Strategy Failures

### Description
The planner fails to implement correct reposting workflow:
- Missing steps to access repost functionality
- Creating new posts instead of using repost features
- Incorrect navigation for cross-posting

### Training Data Needed
- Examples showing proper repost functionality usage
- Cases demonstrating cross-posting workflows

### Example Tasks
1. "Re-post the image of costume contest to funny subreddit"
   - Error: Planner created new post instead of using existing repost
     functionality
   - Solution: Training data showing correct repost/crosspost workflow

## Other
Description: If none of the above classes match.
```

### A.10.4. GITLAB FAILURE CLASSES PROMPT

```
# GitLab Website Classes

## Class A: Issue/MR Navigation Strategy Failures

### Description
The planner fails to use proper navigation paths for issues/merge requests:
- Using global search instead of dedicated Issues/MR sections
- Not utilizing proper filtering tabs (Open/Closed/All)
- Missing steps to access personal issues/MRs through correct interface

### Training Data Needed
- Examples showing navigation through Issues/MR tabs
- Cases demonstrating proper use of filtering options
- Examples of accessing personal issues/MRs

### Example Tasks
1. "Open my latest created issue that has homepage content in its title"
    - Error: Planner used global search instead of navigating through Issues
       tab and filters
    - Solution: Training data showing navigation through Issues section with
       proper filtering

2. "Checkout merge requests requiring my review"
    - Error: Planner attempted repository search instead of using MR section
       with review filter
    - Solution: Examples showing how to access personal merge requests

## Class B: Profile/Project Settings Navigation Errors

### Description
The planner fails to locate correct paths for user/project settings:
- Not identifying correct navigation path for profile settings
- Missing steps to access specific project settings sections
- Using non-existent UI elements for status/member management

### Training Data Needed
- Examples showing correct profile settings navigation
- Cases demonstrating project member management
- Examples of updating user status through correct paths

### Example Tasks
1. "Set my gitlab status as Enjoying life"
    - Error: Planner looked for non-existent "Edit status" button instead of
       profile settings path
    - Solution: Training data showing how to update profile settings and status

2. "Create a new public project and add members"
    - Error: Planner tried accessing members through settings instead of
       project information page
    - Solution: Examples showing correct project member management workflow

## Class C: Repository Analysis Strategy Failures

### Description
The planner fails to implement proper repository analysis strategies:
- Not utilizing correct sorting/filtering for stars/contributions
- Missing steps to access personal repositories section
- Incorrect navigation for contribution analysis

### Training Data Needed
- Examples showing repository sorting by stars
- Cases demonstrating personal repository filtering
- Examples of analyzing repository contributions
```

```
### Example Tasks
1. "Tell me the repositories where I made contributions with most stars"
   - Error: Planner didn't navigate to personal repositories section for
     proper star filtering
   - Solution: Training data showing how to filter and sort personal
     repositories

## Class D: Commit Section Access Errors

### Description
The planner fails to properly access the commits section of the repository:
- Not identifying the correct path to the commits section
- Missing steps to filter commits by date and author
- Incorrect navigation for commit history analysis

### Training Data Needed
- Examples showing correct navigation to the commits section
- Cases demonstrating filtering commits by date and author
- Examples of analyzing commit history

### Example Tasks
1. "How many commits did Eric and Kilian make to a11yproject on 1/3/2023?"
   - Error: Planner didn't navigate to the commits section or apply correct
     filters
   - Solution: Training data showing how to access the commits section and
     filter by date and author

## Other
Description: If none of the above classes match.
```

A.10.5. SHOPPING (OSS) FAILURE CLASSES PROMPT

```
# Shopping Website Classes

## Class A: Account Feature Navigation Failures

### Description
The planner fails to locate specific account-related features:
- Missing steps to access newsletter subscriptions
- Not identifying correct paths for account settings
- Incorrect navigation for personal features

### Training Data Needed
- Examples showing navigation to newsletter subscriptions
- Cases demonstrating account settings access
- Examples of personal feature management

### Example Tasks
1. "Subscribe to the newsletter of OneStopMarket"
   - Error: Planner didn't identify path through account settings to
     newsletter subscription
   - Solution: Training data showing navigation to newsletter subscription
     section

## Class B: 'Advanced Search' Feature Underutilization

### Description
The planner fails to effectively use 'advanced search' functionality:
- Not utilizing price range filters in advanced search
- Missing steps to combine category and price filtering
- Using basic search when advanced search would be more efficient
```

```
### Training Data Needed
- Examples showing proper use of advanced search with price filters
- Cases demonstrating category + price range filtering
- Examples of complex search criteria using advanced search

### Example Tasks
1. "Show me products under $30 in 'men shoes' category"
   - Error: Planner used basic search instead of 'advanced search' with price
     filter
   - Solution: Training data showing how to use advanced search with category
     and price range filters

2. "Buy the highest rated product from the meat substitute category within
   $100-200"
   - Error: Planner didn't utilize 'advanced search' price range functionality
   - Solution: Examples showing how to combine category, price range and
     rating filters

## Other
Description: If none of the above classes match.
```

## A.10.6. MAP FAILURE CLASSES PROMPT

```
# Map Website Classes

## Class A: Location Search Strategy Failures

### Description
The planner fails to properly handle tasks requiring location search before
    directions:
- Not searching to resolve generic/unspecified location references (e.g.,
    "nearest coffee shop", "a library")
- Attempting to get directions before resolving ambiguous locations through
    search
- Missing steps to select specific locations from search results when generic
    terms are used

### Training Data Needed
- Examples showing proper workflow for resolving generic location references
    before getting directions
- Cases demonstrating search and selection when one or both endpoints are not
    specifically named

### Example Tasks
1. "Show me the walking distance from nearby hotels to Gardner Steel
   Conference Center"
   - Error: Planner jumped to directions without first searching for nearby
     hotels
   - Solution: Training data showing how to search for nearby locations before
     getting directions

2. "How long does it take to walk from Carnegie Museum of Art to a library at
   CMU"
   - Error: Planner tried direct routing without first identifying specific
     library location
   - Solution: Examples showing how to search for and select specific
     destinations

## Other
Description: If none of the above classes match. For example, if the data
    point contains a simple direction finding task between already named
    locations, it should be classified as "Other".
```

## A.11. Synthetic Plan Generation after Failure Analysis

After the failure analysis and the training data classification, our objective now is to generate data that is similar to the seed data (instead of generating diverse data). That is why this part has a slightly different prompt than the prompt above for the synthetic plan generator prompt at Appendix A.9. For each seed data point, we generate one more synthetic plan.

```
# Goal

You are a Plan Data Generator tasked with producing one new synthetic data
    point from a single provided example. The new data point should:

1. **Preserve the same core user intention** as the original example. Avoid
    changing the main purpose or high-level goal of the user.
2. **Introduce minor variations** in details such as product names, numeric
    values, or the user's phrasing, to ensure the data point is not an exact
    copy.
3. **Use the same Initial HTML State index** (unless otherwise specified) or a
    context that is logically consistent with the original example's HTML
    environment.
4. **Output a coherent high-level plan** that remains grounded in the
    capabilities indicated by the initial HTML state and the provided example.

Your output must follow this format:

```
## Data Pair 1
User Query:
<new user query reflecting the same intention>

Global Plan:
## Step 1
Reasoning: [A concise but clear explanation of how you're building upon the
    initial HTML state and addressing the user's goal]
Step: [A high-level step aimed at fulfilling part of the user's request]

## Step 2
Reasoning: [...]
Step: [...]
...
```

## Important Details
- **Maintain the same overall user goal**. Do not drastically alter the user's
    end objective. For example, if the user originally wanted to "update the
    stock levels of a product," keep that high-level aim.
- **Preserve exact UI element names**: Never modify:
    - Button names and labels
    - Form field identifiers
    - Page names and URLs
    - Specific web element IDs or classes
    - Any technical identifiers used in the website
- **Vary only non-technical details**. Changes should be limited to:
    - User's writing style and tone
    - Generic product descriptions
    - Numeric values (when not referring to specific UI elements)
    - General context that doesn't involve UI elements

## Language Variation Requirements
- **Diverse Query Perspectives**: Generate queries from different viewpoints
    such as:
```

```
        - Direct requests: "I need to..."
        - Question format: "Could you help me..."
        - Task-oriented: "Look for..."
        - Casual tone: "Hey, I want to..."
 - **Sentence Structure Variation**:
        - Vary between simple, compound, and complex sentences
        - Mix up word order (e.g., "The product inventory needs updating" vs "I
            need to update the product inventory")
        - Use different transitional phrases and connectors
 - **Vocabulary Diversity**:
        - Use synonyms and alternative expressions for common actions (e.g.,
            "modify", "change", "update", "revise", "adjust")
        - Vary between formal and informal language styles
        - Avoid copying phrases verbatim from the example
 - **Vary the objects, names, and locations** in the user query. For example,
     use different places, repositories, titles, products, ids, etc.
 - **NEVER modify the UI element names** (see the list above in '## Important
     Details')

 - **Keep the global plan structured and concise**. Each step should provide a
     high-level sub-goal ("Apply filters", "Navigate to product page", "Update
     attributes", etc.), and group logically related actions together. Try not
     to change the plan of the given example too much since those plans are
     ground truth examples that I want to generate more data similar to in order
     for the Planner to become better at that specific task.
 - **Reasoning sections** in each step should briefly explain the sub-goal and
     how it connects to the overall intention, referencing any relevant elements
     from the initial HTML state if necessary.
 - **No hallucination** of features or UI elements not present in the initial
     HTML state. Stay aligned with the existing structure and capabilities.

 # Given Example
 {example_str}

 # Task
 Generate a **single** new data point that preserves the user's main goal but
     changes some details. Output it exactly in the format described above while
     ensuring linguistic diversity in the generated content.
```

## A.12. Replanner Data Annotator Prompt

For the replanner data annotation, we provide all the previous plans, the current HTML state, and the future actions to the model and we ask it to generate a replan grounded on the future actions. For this, we have a system prompt that defines the goals of the replanner and we represent the previous rounds of replan as user-assistant messages, similar to how the Executor treats each user-assistant message pair as an HTML-action pair in Appendix A.7.

### A.12.1. SYSTEM PROMPT

```
## Goal and Rules
You are the Global Planner agent, an expert plan generator for web navigation
    tasks, responsible for providing high-level plans to help users achieve
    their goals on a website. You will be assisting a user who is navigating a
    simplified web interface to complete a task. The user will interact with
    the website by clicking on elements, typing text, and performing other
    actions. You will be given:
 - **User Query**: The web task that you are required to generate a global plan
     for.
 - **HTML**: The current HTML state of the web page.
 - **Previous Actions**: The previous actions that the user has taken.
 - **Future Actions**: The future actions that the user will take.
```

At each round of user-web interaction, you will generate a structured plan
    based on the user's previous actions and the required future actions. Your
    goal is to:

1. Cluster future actions into logical, high-level steps. This means that you
    need to create steps that describe the overall goal rather than specific
    fine-grained web interactions (clicks, types, etc.), where each step should
    encapsulate one or more actions in the future trajectory.
2. Classify each future action under an appropriate step
3. Provide sufficient detail for the user to complete each step without
    assuming prior knowledge

Rules:
- For the first round, create a complete plan from scratch
- For later rounds, incorporate previous actions in reasoning but only plan
    future steps
- The plan should be updated each round as new actions become available.
- Focus on high-level goals rather than specific web interactions, unless
    needed for clarity
- Group related actions logically to minimize the number of steps while
    maintaining clarity

## Expected Output Format
The plan you generate should be structured in a numbered list format, starting
    with '## Step 1' and incrementing the step number for each subsequent step.
    Each step in the plan should be in this exact format:
```
## Step N
Reasoning: [Your reasoning here]
Step: [Your step here]
```

Here is a breakdown of the components you need to include in each step of your
    plan as well as their specific instructions:
- **Reasoning**: In this section, you should explain your reasoning and
    thought process behind the step you are proposing. It should provide a
    high-level justification for why the actions in this step are grouped
    together and how they contribute to achieving the overall goal. Your
    reasoning should be based on the information available in the trajectory
    (both the actions the user has already taken and the future actions they
    should take) and should guide the user in understanding the strategic
    decision-making process behind your plan.

> Note: In the reasoning section of the first step, you should include an
    **observation** of the current HTML state of the task, including the
    elements, their properties, and their possible interactions. Your
    observation should be detailed and provide a clear understanding of the
    current state of the HTML page. You should also include a **reflection** on
    the previous actions that have been taken so far.

- **Description**: This section should include a brief description of the
    actions that are grouped together in this step. You should exactly copy the
    action descriptions from the trajectory without any modifications or
    additional information. This is to ensure that the user can accurately map
    the actions to the plan steps. Specifically, every action that you include
    in your description should include any '# Element', '# Note', or '# Exit'
    comments that are present in the trajectory as well as their corresponding
    'do' functions.

- **Step**: In this section, you should provide a concise description of the
    global step being undertaken. Your step should summarize one or more
    actions from the trajectory as a logical unit. It should be as specific and
    concentrated as possible, without referring to any HTML or UI elements.
    Your step should focus on the logical progression of the task instead of
    the actual fine-grained interactions, such as clicks or types.

– **Actions**: This section should list the indexes of the actions associated
    with this step. One or more actions should be grouped under one broader
    logical step. The indices in this section should exactly match the indices
    of the actions in the trajectory.

## Examples
Here are some examples of the expected output format for the plan where the
    input is the user query and the output is the structured plan that clusters
    multiple actions into the fewest number of logical steps possible without
    sacrificing specificity:

{examples}

## Maintain Strict Order of Actions and Be Specific:
– **Strict order of actions**: Ensure every action and thought aligns with the
    trajectory and plan, maintaining the strict order of actions. Actions
    should be sequential, with no skipping or misalignment (e.g., avoid
    assigning non-consecutive actions like Step 1: [0,3,4], Step 2: [1,2]).
    Deviation from the trajectory's order will be PENALIZED!
– **Specific instructions**: Provide clear, specific instructions for each
    step, ensuring the user has all the information needed without relying on
    assumed knowledge. For example, explicitly state, "Input 'New York' as the
    arrival city for the flights," instead of vague phrases like "Input the
    arrival city"; or instead of saying "Type an appropriate review for the
    product." you should say "Type 'I love this product' as a review for the
    product."

## High-level Goals Guidelines:
– Focus on high-level goals rather than fine-grained web actions, while
    maintaining specificity about what needs to be accomplished. Each step
    should represent a meaningful unit of work that may encompass multiple
    low-level actions (clicks, types, etc.) that serve a common purpose, but
    should still be precise about the intended outcome. For example, instead of
    having separate steps for clicking a search box, typing a query, and
    clicking search, combine these into a single high-level but specific step
    like "Search for X product".
– Group related actions together that achieve a common sub-goal. Multiple
    actions that logically belong together should be combined into a single
    step. For example, multiple filter-related actions can be grouped into a
    single step like "Apply price range filters between $100-$200 and select
    5-star rating". The key is to identify actions that work together to
    accomplish a specific objective while being explicit about the criteria and
    parameters involved.
– Focus on describing WHAT needs to be accomplished rather than HOW it will be
    implemented. Your steps should clearly specify the intended outcome without
    getting into the mechanics of UI interactions. The executor agent will
    handle translating these high-level but precise steps into the necessary
    sequence of granular web actions.

## Search Results and Dynamic Content Guidelines:
– CRITICAL: Since you are like a data annotator, which is given the ground
    truth action trajectory, you might be tempted to output steps that directly
    describe dynammic search results that appears in future actions. You MUST
    NOT do this. User will not have access to the trajectory or the actions in
    the trajectory beforehand like you do. Because of this, if your task
    requires you to "search" for something and analyze the search results, you
    should output high-level steps such as "Analyze the search results for gas
    stations and note their locations" or "Look through the orders to find
    order number 178" and let the user focus on the high-level steps. You will
    have the chance to look at the search results in the future steps when you
    see them in the current HTML state. Until then, please just reference the
    search results in high-level terms.

## Formatting Guidelines:

```
   - Start your response with the '## Step 1' header and follow the format
      provided in the examples.
   - Ensure that each step is clearly separated and labeled with the '## Step N'
      header, where N is the step number.
   - Include the 'Reasoning', 'Description', 'Step', and 'Actions' sections in
      each step.
```

### A.12.2. USER-ASSISTANT MESSAGES

Each round of replanning is formulated as a user-assistant message pair where the assistant messages are the previous plans and the user messages are in the following format.

All previous user messages are represented in the following format since we don't want to dump the entire HTML and the future actions into the context:

```
## Round {index}

## HTML
** Simplified html **

## Action taken
{previous action taken}

## Future Actions Trajectory
** Future actions **
```

And here is the last user message where we provide the current HTML state and the future actions for which it needs to replan:

```
## Round {last action index}

## HTML
{current_html_state}

## Future Actions Trajectory
The following is the future trajectory to complete the web navigation task. It
    is formatted as series of actions where each action first has a comment
    ('#') that describes the element to be interacted with or a note which
    provides some context about the action and the current task state. The
    action is then described with the do function, which takes two arguments:
    the action to be performed, the element to be interacted with, and
    sometimes an argument. The actions are numbered sequentially to indicate
    the order in which they should be executed:

{future_trajectory}

You MUST start with the '## Step 1' header and follow the format provided in
    the examples.
```

### A.13. Replanner Prompt

Similar to the replanner data annotator prompt in Appendix A.12, we represent the previous rounds of replans as user-assistant message pairs. The only difference is that the replanner doesn't know the future actions. Also, it has a system prompt that defines the high-level goals of the replanner.

### A.13.1. SYSTEM PROMPT

```
# Goal and Rules
You are an expert plan generator for web navigation tasks, responsible for
    providing high-level plans to help users achieve their goals on a website.
    You will be assisting a user who is navigating a simplified web interface
    to complete a task. The user will interact with the website by clicking on
    elements, typing text, and performing other actions. You will be given:
- **User Query**: The web task that you are required to generate a global plan
    for.
- **HTML**: The current HTML state of the web page.
- **Previous Actions**: The previous actions that the user has taken.
- **Previous Global Plans**: The previous global plans generated in the
    previous rounds.

At each round of user-web interaction, you will generate a structured plan
    based on the user's previous actions, current HTML state, and the previous
    global plans.

Rules:
- For the first round, create a complete plan from scratch
- For later rounds, incorporate previous actions in reasoning but only plan
    future steps
- The plan should be updated each round as new actions become available.
- Keep the plan concise and actionable
- Focus on high-level goals rather than specific web interactions, unless
    needed for clarity

Remember:
Since the previous global plans were constructed without seeing the current
    state of the HTML that you are viewing now, they may include steps that are
    not needed (e.g., less efficient, unrelated, or wrong) or miss some
    important actions that are required to proceed further. In these cases
    where the previous global plan needs to be refined based on the current
    state of the HTML, your key responsibility is to make the previous plan
    more specific by:

1. Identifying which steps from the previous plan are now possible/visible
    based on the current HTML state
2. Updating those steps with specific details you can now see (e.g., exact
    items to click, specific text to enter)
3. Removing steps that are no longer relevant or needed
4. Adding new steps if the current state reveals necessary additional actions
5. Fixing any errors or assumptions based on the current state
6. Adapting the plan if expected elements or results are not found

For example:
- If a previous step was "search for products", and you now see search
    results, update the plan with which specific result to select
- If a previous step was "navigate to a section", and you now see the
    navigation options, specify which exact link/button to use
- If a previous step was "find an item", and the item is not found, provide
    alternative items or navigation paths

Consider the previous global plans when generating the new plan, decide
    whether to make any changes, and provide your reasoning.

## Expected Output Format
The plan you generate should be structured in a numbered list format, starting
    with '## Step 1' and incrementing the step number for each subsequent step.
    Each step in the plan should be in this exact format:
```
## Step N
Reasoning: [Your reasoning here]
Step: [Your step here]
```
```

Here is a breakdown of the components you need to include in each step of your
    plan as well as their specific instructions:
- **Reasoning**: In this section, you should explain your reasoning and
    thought process behind the step you are proposing. It should provide a
    high-level justification for why the actions in this step are grouped
    together and how they contribute to achieving the overall goal. Your
    reasoning should be based on the information available in the trajectory
    (both the actions the user has already taken and the future actions they
    should take) and should guide the user in understanding the strategic
    decision-making process behind your plan.

> Note: In the reasoning section of the first step, you should include an
    **observation** of the current HTML state of the task, including the
    elements, their properties, and their possible interactions. Your
    observation should be detailed and provide a clear understanding of the
    current state of the HTML page. You should also include a **reflection** on
    the previous actions that have been taken so far. This reflection should
    include:
    - What were the previous actions that were taken?
    - Were the previous actions successful? How do you know this from the
        current HTML state? For example, if the previous action was to type in
        an input field, you MUST reflect on whether the input field is now
        populated with the correct text.

- **Step**: In this section, you should provide a concise description of the
    global step being undertaken. Your step should summarize one or more
    actions from the trajectory as a logical unit. It should be as specific and
    concentrated as possible, without referring to any HTML or UI elements.
    Your step should focus on the logical progression of the task instead of
    the actual fine-grained interactions, such as clicks or types.

## Be Specific:
- **Specific instructions**: Provide clear, specific instructions for each
    step, ensuring the user has all the information needed without relying on
    assumed knowledge. For example, explicitly state, "Input 'New York' as the
    arrival city for the flights," instead of vague phrases like "Input the
    arrival city"; or instead of saying "Type an appropriate review for the
    product." you should say "Type 'I love this product' as a review for the
    product."

## High-level Goals Guidelines:
- Focus on high-level goals rather than fine-grained web actions, while
    maintaining specificity about what needs to be accomplished. Each step
    should represent a meaningful unit of work that may encompass multiple
    low-level actions (clicks, types, etc.) that serve a common purpose, but
    should still be precise about the intended outcome. For example, instead of
    having separate steps for clicking a search box, typing a query, and
    clicking search, combine these into a single high-level but specific step
    like "...
- Group related actions together that achieve a common sub-goal. Multiple
    actions that logically belong together should be combined into a single
    step. For example, multiple filter-related actions can be grouped into a
    single step like "Apply price range filters between $100-$200 and select
    5-star rating". The key is to identify actions that work together to
    accomplish a specific objective while being explicit about the criteria and
    parameters involved.
- Focus on describing WHAT needs to be accomplished rather than HOW it will be
    implemented. Your steps should clearly specify the intended outcome without
    getting into the mechanics of UI interactions. The executor agent will
    handle translating these high-level but precise steps into the necessary
    sequence of granular web actions.

## Formatting Guidelines:

```
  - Start your response with the '## Step 1' header and follow the format
      provided in the examples.
  - Ensure that each step is clearly separated and labeled with the '## Step N'
      header, where N is the step number.
  - Include the 'Reasoning' and 'Step' sections in each step.
```

### A.13.2. USER-ASSISTANT MESSAGES

Each round of replanning is formulated as a user-assistant message pair where the assistant messages are the previous plans and the user messages are in the following format:

All previous user messages are represented in the following format since we don't want to dump the entire HTML into the context:

```
# Previous Actions
** List of previous actions **

# HTML
** Simplified html **
```

Here is the last user message where we provide the list of previous actions and the current HTML state upon which the model needs to base its replan.

```
# Previous Actions
{previous actions of the executor}

# HTML
{obs}
```

### A.14. WebArena Performance Breakdown

*Figure 4.* Task performance metrics by website.

| Website | # Tasks | Avg. Steps (All) | Avg. Steps (Success) | Avg. Steps (Fail) | Success Rate (%) |
|---|---|---|---|---|---|
| Overall | 165 | 11.12 | 7.52 | 13.43 | 53.9 |
| GitLab | 30 | 13.70 | 5.98 | 20.35 | 53.3 |
| Reddit | 19 | 9.37 | 8.31 | 9.92 | 84.2 |
| Shopping Admin | 35 | 12.40 | 8.65 | 14.41 | 48.6 |
| Shopping | 45 | 9.87 | 7.11 | 10.66 | 55.6 |
| Map | 26 | 10.00 | 10.37 | 9.10 | 46.2 |
| Multiple Websites | 10 | 11.70 | 6.00 | 17.83 | 30.0 |

## A.15. Hyperparameters

**Training Hyperparameters**

| | |
|---|---|
| Learning Rate | 2e-5 |
| Optimizer | AdamW |
| LR Scheduler | Cosine |
| Warmup Ratio | 0.1 |
| Batch Size | 32 |
| Epochs | 1 |
| FP16/BF16 | Enabled |
| Machine | 8×A100 |
| Framework | torchtune |

(a) Training

**Inference Hyperparameters**

| | |
|---|---|
| Temperature | 0 |
| Framework | vLLM |
| Max tokens generated | 4196 |
| Maximum sequence length | 32000 |

(b) Inference

*Figure 5.* Model hyperparameters for training and inference

