# OpenReview forum: "Plan-and-Act: Improving Planning of Agents for Long-Horizon Tasks"
_ICML.cc/2025/Conference — ICML 2025 poster_

### Official Review · Reviewer_FBh4 · 2025-03-09

**Overall Recommendation:** 3

**Summary:**

The paper proposes a Plan-and-Act methodology for long-horizon web-tasks. The basic premise of the Plan-and-Act method is that it decomposes the long-horizon planning into two modules: planning and executing. The planner module creates a long-horizon plan and the executor executes actions relevant to completing the tasks in this plan. The paper also introduces a dataset augmentation and synthesis process to fine-tune LLMs in the planning phase. The model achieves a ~10% increase in success rates compared to previous SOTA in the Web-Arena Lite benchmark.

**Claims And Evidence:**

The claims made in the paper are supported well with the experiments. But that being said, I am always a bit skeptical about papers that operate with LLMs for evaluations wrt to reproducibility. The experimental process for reproducing the tables is not provided in the paper (seeds, temperature, etc.). This makes it slightly harder to judge the effectiveness of the paper as I am not sure if the best results obtained with the LLMs were included in the paper. I would urge the authors to give more information on the reproducibility of the tables in the paper and any experimental parameters so that it is easier for the reader to reproduce.

**Essential References Not Discussed:**

The papers below operate with a similar plan-act (+extra modules) for embodied robotics tasks. My initial thought was that something like this could be extended to the WebArena environment as well.
[1]: Chan Hee Song, Jiaman Wu, Clayton Washington, Brian M. Sadler, Wei-Lun Chao, and Yu Su. LLMPlanner: Few-Shot Grounded Planning for Embodied Agents with Large Language Models, 2023
[2]: Nayak, S., Morrison Orozco, A., Have, M., Zhang, J., Thirumalai, V., Chen, D., ... & Balakrishnan, H. (2024). Long-horizon planning for multi-agent robots in partially observable environments. Advances in Neural Information Processing Systems, 37, 67929-67967.
[3]: Shyam Sundar Kannan, Vishnunandan LN Venkatesh, and Byung-Cheol Min. SMART-LLM: Smart Multi-Agent Robot Task Planning using Large Language Models. arXiv preprint arXiv:2309.10062, 2023

**Experimental Designs Or Analyses:**

The experimental design is valid as it is based on a benchmark dataset (WebArena-Lite). I would suggest the authors to include experimental details for better reproducibility of the experiments.

**Methods And Evaluation Criteria:**

I feel that the method introduced in this work has been introduced in a few other research papers (refer essential references not discussed section) before as well, albeit for other applications (robotics, etc.) These papers also focus on decomposing the long-horizon planning into multiple modules (planning + acting w/ replanning). I would urge the authors to discuss the differences of these prior works when compared to Plan-and-Act and maybe show a comparison against these methods as well. It might be possible that the superior performance might be due to the long customised prompts given as input to the LLMs (as shown in the appendix).

Right now, in its current state, I am going to lean towards weak accept but am willing to change if the authors can address some of the concerns I have raised in my review (specifically, the comparison to other plan-act-based methods)

The evaluation criteria seem to be from a standard benchmark and I have no qualms about it. The benchmark is apt for the application at hand and the proposed method performs better than previous SoTA.

**Other Comments Or Suggestions:**

NA

### Update after rebuttal:
I wasn't sure if the "official comments" were visible to the authors and hence am including them here:

I appreciate the authors' rebuttal. I am assuming that the authors will include the hyperparameters, discussion of additional experiments on WebVoyager, other missing references raised in the reviews in the camera-ready. I am increasing my score by 1 as the authors have clarified my questions.

A suggestion: it would be nicer for the reviewers if you could copy the text to particular answers to questions raised in the rebuttal instead of redirecting to rebuttals of other reviewers. It gets hard to do that in the current state eg. "please see R3-5 for more details".

**Other Strengths And Weaknesses:**

I appreciate the way in which the results are presented where the increase in success rates is clearly visible with the addition of each module.

**Questions For Authors:**

I would urge the authors to present more experimental details for better reproducibility and a thorough explanation on why Plan-and-Act is different than other plan-act methods used in the papers linked above.

**Relation To Broader Scientific Literature:**

The results show improvement compared to previous SoTA. But the method used has been introduced previously for other applications. I am not sure if this would count as a novel contribution to the scientific literature.

**Theoretical Claims:**

No theoretical claims in the paper. Mostly experimental.

---

> ### Author Rebuttal · Authors · 2025-04-01
>
> > R4-1: The experimental process for reproducing the tables is not provided in the paper (seeds, temperature, etc.). This makes it slightly harder to judge the effectiveness of the paper as I am not sure if the best results obtained with the LLMs were included in the paper. I would urge the authors to give more information on the reproducibility of the tables in the paper and any experimental parameters so that it is easier for the reader to reproduce.
>
> Hyperparameters for SFT
>
> We have used the following parameters for training both the Planner and the Executor models, both for the 70B and 8B models. As the 70B model, we have used Llama-3.3-70B-Instruct as the base model; and for the 8B model, we have used Llama-3.1-8B-Instruct as the base model.
>
>
> - Learning Rate: 2e-5
> - Optimizer: AdamW
> - LR Scheduler: Cosine
> - Warmup Ratio: 0.1
> - Batch Size: 32
> - Epochs: 1
> - FP16/BF16: Enabled
> - Machine: 8xA100
> - Framework: torchtune
>
> Hyperparameters For Inference
>
> - Temperature: 0
> - Framework: vLLM
> - Max tokens generated: 4196
> - Maximum sequence length: 32000
>
> Hyperparameters for Data Generation
>
> We used GPT-4o for all data generation stages.
> When generating synthetic data, for each generation, we have retrieved 5 in-context examples and generated 10 new synthetic user query-plan pairs.
>
>
>
> > R4-2: I feel that the method introduced in this work has been introduced in a few other research papers (refer essential references not discussed section) before as well, albeit for other applications (robotics, etc.) These papers also focus on decomposing the long-horizon planning into multiple modules (planning + acting w/ replanning). I would urge the authors to discuss the differences of these prior works when compared to Plan-and-Act and maybe show a comparison against these methods as well. It might be possible that the superior performance might be due to the long customised prompts given as input to the LLMs (as shown in the appendix).
>
> We thank the reviewer for their feedback. For method distinctions, see R3-4 in our response to Reviewer 3. For comparisons on other datasets, please see R2-1 (WebVoyager)/R2-2 (WebArena) in our response to Reviewer 2, where we show that our model performs on-par with other prior work on WebArena and set a new SOTA for text-only models on WebVoyager.
>
> Regarding the prompts given as input to the LLMs, most of the customized prompts in the appendix are for the data generation pipeline. At inference time, the System prompts for the Planner, Executor, and for Replanning are only the prompts in A.3.1, A.4.1, and A.10.1, which are fairly high-level and generic and similar in length to other prior work (See Figure 21 in WebRL).
>
>
> > R4-3: The experimental design is valid as it is based on a benchmark dataset (WebArena-Lite). I would suggest the authors to include experimental details for better reproducibility of the experiments.
>
> Please see R4-1.
>
>
> > R4-4: The results show improvement compared to previous SoTA. But the method used has been introduced previously for other applications. I am not sure if this would count as a novel contribution to the scientific literature.
>
> Please see R3-4/R4-5.
>
>
> > R4-5: The papers below operate with a similar plan-act (+extra modules) for embodied robotics tasks. My initial thought was that something like this could be extended to the WebArena environment as well. [1]: Chan Hee Song, Jiaman Wu, Clayton Washington, Brian M. Sadler, Wei-Lun Chao, and Yu Su. LLMPlanner: Few-Shot Grounded Planning for Embodied Agents with Large Language Models, 2023 [2]: Nayak, S., Morrison Orozco, A., Have, M., Zhang, J., Thirumalai, V., Chen, D., ... & Balakrishnan, H. (2024). Long-horizon planning for multi-agent robots in partially observable environments. Advances in Neural Information Processing Systems, 37, 67929-67967. [3]: Shyam Sundar Kannan, Vishnunandan LN Venkatesh, and Byung-Cheol Min. SMART-LLM: Smart Multi-Agent Robot Task Planning using Large Language Models. arXiv preprint arXiv:2309.10062, 2023
>
> We thank the reviewer for these additional references which we will add to our related work. Indeed, these works [1,2,3] use hierarchical LLM Agents to decompose tasks and plan for robotics/embodied agents which shares some similarity with Plan-and-Act. However, similar to the other planning based web agents mentioned in R3-4, none of these prior work contain a framework for collecting and generating synthetic data for training open source LLMs to get better at these tasks. The synthetic data generation pipeline of Plan-and-Act is what differentiates it from other prior work with planning and agents.
>
>
> > R4-6: I would urge the authors to present more experimental details for better reproducibility and a thorough explanation on why Plan-and-Act is different than other plan-act methods used in the papers linked above.
>
> Please see R4-1 for more experimental details and R3-4/R4-5 for a detailed explanation on how Plan-and-Act is different from other plan-act methods.

---

### Official Review · Reviewer_6X2E · 2025-03-14

**Overall Recommendation:** 3

**Summary:**

This paper proposes Plan-and-Act, an agent for web environments which separates planning from execution. A planner generates the overall plan, and a separate executor carries out the plan by issuing low-level actions. In order to train the planner, a synthetic data generation method is introduces to annotate trajectories with feasible plans. The method achieves a new sota on WebArena-Lite.

**Claims And Evidence:**

The proposed method outperforms the previous sota, WebRL, and well designed ablations show that proposed components in planner and executor design contribute to the performance.

**Essential References Not Discussed:**

There are numerous prior works which propose a dynamic plan-and-execute architecture, which are not discussed in the paper:
- AgentOccam: A Simple Yet Strong Baseline for LLM-Based Web Agents, ICLR 2025
- WebPilot: A Versatile and Autonomous Multi-Agent System for Web Task Execution with Strategic ExplorationIiii
- Adaptive planning from feedback with language models, NeurIPS 2023

**Experimental Designs Or Analyses:**

see above

**Methods And Evaluation Criteria:**

Methods are evaluated on Webarena-lite success rate. Baselines include finetuning ReAct style without a planner, and WebRL.

**Other Comments Or Suggestions:**

- It would be beneficial to report results using a smaller LLama model as well, to demonstrate the generality of the proposed approach.

**Other Strengths And Weaknesses:**

The strengths of the paper are training a strong plan-and-execute approach using synthetic data, while its weaknesses are the lack of discussion of related works and limited novelty, as plan-and-execute is a common approach among related works. I am willing to increase my score if the authors address these points.

**Questions For Authors:**

see above

**Relation To Broader Scientific Literature:**

There are several issues with the paper in terms of relation to the broader literature,
- It is difficult to understand the contributions of the paper within the broader literature of planning in LLM agents, as discussion of related works in agents with planning is missing
- The paper does not discuss the difference between Webarena and Webarena Lite, and it is difficult to understand the distinction of the approach among the many approaches in Webarena

**Theoretical Claims:**

n/a

---

> ### Author Rebuttal · Authors · 2025-04-01
>
> > R3-1: It is difficult to understand the contributions of the paper within the broader literature of planning in LLM agents, as discussion of related works in agents with planning is missing
>
> We thank the reviewer for their feedback. Please see response R3-4.
>
>
> > R3-2: The paper does not discuss the difference between Webarena and Webarena-Lite, and it is difficult to understand the distinction of the approach among the many approaches in Webarena
>
> We appreciate the reviewer’s feedback. WebArena-lite was introduced in VisualAgentBench as a subset of the full WebArena, refined to remove unclear and impossible tasks, and also generated a training set where the original WebArena benchmark does not. For a discussion on how our approach differs from other prior work in this area, please refer to R3-4. For a comparison of results, please refer to R2-2, where we show that Plan-and-Act performs on-par with existing prior work on the full WebArena benchmark while being completely open source.
>
>
> > R3-3: There are numerous prior works which propose a dynamic plan-and-execute architecture, which are not discussed in the paper:
> >
> > AgentOccam: A Simple Yet Strong Baseline for LLM-Based Web Agents, ICLR 2025
> > WebPilot: A Versatile and Autonomous Multi-Agent System for Web Task Execution with Strategic ExplorationIiii
> > Adaptive planning from feedback with language models, NeurIPS 2023
>
>
> We appreciate the feedback from the reviewer. AgentOccam and WebPilot were referenced in the Introduction as well as Section 2.1 of the Related Work section and we thank the reviewer for the pointer to AdaPlanner. We will expand and discuss these works in the final version of the paper. For a more in-depth discussion of how our work differs from the existing literature, please see R3-4.
>
>
> > R3-4:   … its weaknesses are the lack of discussion of related works and limited novelty, as plan-and-execute is a common approach among related works. I am willing to increase my score if the authors address these points.
>
> The Plan-and-Act framework is not just a hierarchical planning framework, but also a data generation framework. Prior work that involve planning and hierarchies such as AgentOccam, WebPilot, AdaPlanner, and ADaPT are all prompting methods using closed-source models such as GPT-4o as their base model. Our method provides a simple, systematic way to generate high quality training data to train LLMs on web tasks.
>
> In addition, our method uses a very simple 2-agent framework, which is significantly simpler compared to other prior work with planning. AgentOccam uses a “Planning via Generation” technique where the planning is incorporated into the action space and the model plans in a tree-like fashion. WebPilot has a significantly more complex infrastructure with 6 different agents in total. AdaPlanner has a In-Plan and Out-of-Plan Refiner to facilitate replanning when the plan is wrong and a skill-discovery module that is orthogonal to our method and can be used in conjunction. ADaPT uses recursive decomposition to decomposes tasks when the executor fails, whereas our dual-agent architecture simply replans at each step. All of these methods use more excessive prompting to improve performance, while our method has a simple Plan-and-Act structure at runtime.
>
> Other work that have discussed generating training data for Web Agents such as DigiRL, WebRL, AutoWebGLM, and NNetNav provide more complex techniques for collecting diverse trajectories, which are complementary to Plan-and-Act, as our pipeline (Section 4.1) is simple and can be interchanged. Furthermore, they only produce trajectory data, but do not planning data (Section 4.2). They also rely on external simulators to generate data, whereas our method can generate synthetic planning data without a simulator (Section 4.3).
>
>
>
> > R3-5: It would be beneficial to report results using a smaller LLama model as well, to demonstrate the generality of the proposed approach.
>
> We thank the reviewer for this feedback. We have also trained a Llama-3.1-8b-instruct model using the Plan-and-Act framework.
>
> We also decided to add CoT style reasoning to the Planner and Executor so that it generates some reasoning before it generates the plan/action.  Furthermore, we also finetuned another Llama-70B model on this data as well to see how the performance would be for the Llama-70B model with CoT as well:
>
>
> - Llama-70B with Dynamic Replanning: 53.94% (89/165)
> - Llama-8B with CoT with Dynamic Replanning: 53.33% (88/165)
> - Llama-70B with CoT with Dynamic Replanning: **57.58**% (95/165)
>
> We evaluated smaller Llama-8B models and found that our approach still outperforms existing methods (53.33%), demonstrating strong generality. With CoT reasoning, our Llama-70B achieves **57.58**%, setting a new SOTA on the WebArena-lite dataset. We will include these results in the final version of our paper.

---

### Official Review · Reviewer_kftV · 2025-03-14

**Overall Recommendation:** 3

**Summary:**

The authors propose Plan-and-Act, which consists of two separate modules for planning and acting (execution), with dynamic replanning for better adaption to different situations. The Planner generates high-level plans, which are taken as input for the Executor to generate low-level actions. Importantly, for the training of the two modules with enough domain knowledge, the authors use synthetic data generation. While the low-level actions can be the training data for the Executor, the authors suggest two approaches for enhanced training of the Planner. The first approach is plan annotation, which annotates collected trajectories with plan "labels" by prompting LLMs. To scale up the training data for Planner, the second approach to synthetically generate plan data is used. On WebArena-Lite, they show that Plan-and-Act can outperform the baselines.

## update after rebuttal

I appreciate the authors for providing the extended empirical results.

While the added empirical results do add value to the work and address some of my concerns, I still believe none of WebArena and WebVoyager justify the claim of being "long-horizon." From my experience, neither WebVoyager nor WebArena requires long-horizon executions. On a relatively minor but related topic, the number of steps for the failure trajectories shouldn't be used for measuring the complexity of tasks (and even successful trajectories can contain non-optimal actions and steps).

**Claims And Evidence:**

- The claims that constitute the proposed approach are empirically supported by the results on WebArena-Lite. Especially, Table 1 provides a performance improvement breakdown across the components of the proposed method.

**Essential References Not Discussed:**

- More papers can be cited in the context of using separate planner and executor, especially "ADaPT: As-Needed Decomposition and Planning with Language Models"

**Experimental Designs Or Analyses:**

- One primary weakness of this work is its empirical evaluation. It only provides the evaluation on WebArena-Lite, which employs non-real-world websites as part of the environment. The experimental results may be strengthened by evaluating the proposed approach on more realistic benchmarks, such as WebVoyager.
- SOTA claims on WebArena-Lite (not just from this submission) may need further investigations, as many papers still use the original WebArena for evaluation and the corresponding performance on the WebArena-Lite subset should be derivable.

**Methods And Evaluation Criteria:**

- The proposed method for synthetic generation of planning data is sound. Especially, for environments or domains where the dynamics itself can change over time, such as web navigation, generating data that is grounded to the actual environment is important.
- (Also as mentioned in this paper,) synthetic augmentation of planning training data can introduce noise to some extent.

**Other Comments Or Suggestions:**

N/A

**Other Strengths And Weaknesses:**

- Given the task distribution of WebArena(-Lite) and example plans and trajectories, the use of the term "long-horizon" can be an overstatement or misleading.

**Questions For Authors:**

N/A

**Relation To Broader Scientific Literature:**

- While the targeted problem itself is relevant to LLM agents, the current empirical evaluation (only on WebArena-Lite) makes it hard to assess the proposed method's broader applicability.

**Theoretical Claims:**

There is not much of theoretical claims from this submission.

---

> ### Author Rebuttal · Authors · 2025-04-01
>
> > R2-1: One primary weakness of this work is its empirical evaluation. It only provides the evaluation on WebArena-Lite, which employs non-real-world websites as part of the environment. The experimental results may be strengthened by evaluating the proposed approach on more realistic benchmarks, such as WebVoyager.
>
> That is a fair point. We have evaluated our method on WebVoyager and report the results below, where we achieve **80.02**% accuracy which is SOTA for text-only models (note that operator uses multi-modal approach ).
>
> To evaluate our approach on WebVoyager, we first collected training data, since WebVoyager does not have any trajectories. We used the text-only WebVoyager model and generated 1500 trajectories using the Action Trajectory Generation (Section 4.1). We then used QWQ-32B to annotate our trajectories (Section 4.2) and to generate 10k synthetic plans (Section 4.3). Our model uses both Dynamic-Replanning and CoT reasoning introduced in R3-5. We finetuned 2 llama-3.1-8b-instruct models for the Planner and Executor.
>
> Furthermore, we tried using QWQ-32B as a zero-shot executor with our finetuned llama-3.1-8b-instruct model as the planner.
>
> Our 8B planner and executor has an accuracy of 58.08% and our 8B planner and 32B executor achieves an accuracy of **80.02**%, which sets a new SOTA for all open source models, as well as one for all text-only models, since OpenAI Operator uses vision.
>
> | Technique              | Base Model                                               | WebVoyager Accuracy (%) |
> | ---------------------- | ------------------------------------------------------- | ----------------------- |
> | WebVoyager (text-only) | gpt-4-turbo                                             | 44.3                    |
> | NNetNav                | llama-8b-instruct                                       | 34.2                    |
> | OpenWebVoyager         | Idefics2-8b-instruct                                    | 27.4                    |
> | Wilbur                 | gpt-4-turbo                                             | 52.6                    |
> | WebVoyager             | gpt-4-turbo                                             | 57.1                    |
> | Plan-and-Act           | llama-8b-instruct planner + llama-8b-instruct executor | 58.08                   |
> | Agent-E                | gpt-4-turbo                                             | 73.1                    |
> | Plan-and-Act           | llama-8b-instruct planner + zero-shot QWQ-32B executor | **80.02**                   |
> | OpenAI Operator        | OpenAI Operator                                         | 87.0                    |
>
>
>
>
> > R2-2: SOTA claims on WebArena-Lite (not just from this submission) may need further investigations, as many papers still use the original WebArena for evaluation and the corresponding performance on the WebArena-Lite subset should be derivable.
>
> We appreciate the feedback from the reviewer. We investigated your suggestion and found that while some papers do release traces that allow you to see the trajectories, the evaluation of these traces is impossible for some tasks without running the simulation itself.
>
> Thus, we evaluated Plan-and-Act on the full WebArena benchmark. We used the Llama 70B model with CoT that we introduced in R3-5. Below, you can see our performance compared to other work prior to the ICML deadline. Plan-and-Act performs better/on-par with all prior work, while being open-source.
>
> | Method           | Base Model      | WebArena Accuracy (%) |
> | ---------------- | --------------- | --------------------- |
> | NNetNav          | Llama-3.1-8b    | 16.3                  |
> | AutoWebGLM       | ChatGLM3-6B     | 18.2                  |
> | WebPilot         | gpt-4o          | 37.2                  |
> | AgentOccam       | GPT-4-Turbo     | 43.1                  |
> | AgentOccam-Judge | GPT-4-Turbo     | 45.7                  |
> | Plan-and-Act     | Llama-70B       | 45.7                  |
> | Openai Operator  | Openai Operator | 58.1                  |
>
>
>
>
> > R2-3: While the targeted problem itself is relevant to LLM agents, the current empirical evaluation (only on WebArena-Lite) makes it hard to assess the proposed method's broader applicability.
>
> Please see R2-1 and R2-2.
>
>
> > R2-4: More papers can be cited in the context of using separate planner and executor, especially "ADaPT: As-Needed Decomposition and Planning with Language Models"
>
> We will add ADaPT in related work.
>
>
> > R2-5: Given the task distribution of WebArena(-Lite) and example plans and trajectories, the use of the term "long-horizon" can be an overstatement or misleading.
>
> We kindly refer the reviewer to the table in R1-1, in the response to Reviewer 1. There, we have provided a breakdown of the number of steps per task in WebArena-lite. Each task averages around 9-13 steps on average.

---

> > ### Comment · Reviewer_kftV · 2025-04-04
> >
> > I appreciate the authors for providing the extended empirical results.
> >
> > While the added empirical results do add value to the work and address some of my concerns, I still believe none of WebArena and WebVoyager justify the claim of being "long-horizon."
> > From my experience, neither WebVoyager nor WebArena requires long-horizon executions.
> > On a relatively minor but related topic, the number of steps for the failure trajectories shouldn't be used for measuring the complexity of tasks (and even successful trajectories can contain non-optimal actions and steps).

---

> > > ### Author Response · Authors · 2025-04-07
> > >
> > > We appreciate the reviewer's valuable feedback and agree that the term 'long-horizon' may lead to misunderstandings given the task distributions in WebArena and WebVoyager. We will clarify this in the revised manuscript by softening the terminology, emphasizing our approach's potential applicability toward longer-horizon tasks, and avoiding overstating current task complexity.

---

### Official Review · Reviewer_4dS1 · 2025-03-16

**Overall Recommendation:** 3

**Summary:**

This paper introduces Plan-and-Act, a framework consisting of a planner that generates high-level task plans and an executor that translates these plans into specific actions. To deal with unexpected failures, the planner will be involved in updating the plan after each execution step. Besides, a synthetic data generation method is proposed to finetune the planner. Through experiments in the WebArena-Lite environment, Plan-and-Act achieves a state-of-the-art success rate.

**Claims And Evidence:**

Yes.

**Essential References Not Discussed:**

No.

**Experimental Designs Or Analyses:**

Yes.

**Methods And Evaluation Criteria:**

Yes, the proposed methods make sense for the web task. However, the paper only reports the success rate of the methods on the WebArena-Lite benchmark. Additional metrics, such as the average number of steps required to complete a task, would provide a more comprehensive assessment.

**Other Comments Or Suggestions:**

No.

**Other Strengths And Weaknesses:**

Strengths:
1. The experimental results are positive, achieving new state-of-the-art performance on the WebArena-Lite benchmark.
2. The paper provides sufficient ablation studies to demonstrate the contribution of individual components.

Weaknesses:
1. The proposed framework lacks novelty, as it essentially follows a hierarchical planning approach and utilizes environmental feedback for replanning—both of which are commonly used in planning applications.
2. The experiments are conducted solely on WebArena-Lite, a simulated environment. It would be more informative to evaluate the approach on WebVoyager, which better reflects real-world web behavior.
3. More prior methods [1, 2] should be included in the experimental comparison to better contextualize the improvements.
4. The writing could be improved in the following aspects:
(1) The Related Work section should not merely summarize previous studies but should also explicitly discuss the similarities and differences between the proposed approach and existing methods.
(2) The paper should include a thorough discussion of the limitations of the proposed method.

References:

[1] Zhang et al., WebPilot: A Versatile and Autonomous Multi-Agent System for Web Task Execution with Strategic Exploration.

[2] Yang et al., AgentOccam: A Simple Yet Strong Baseline for LLM-Based Web Agents.

**Questions For Authors:**

See Weaknesses and Evaluation Criteria.

**Relation To Broader Scientific Literature:**

A new synthetic data generation strategy can improve LLM's performance for long-horizon web tasks.

**Theoretical Claims:**

No, there is no theoretical claim.

---

> ### Author Rebuttal · Authors · 2025-04-01
>
> > R1-1:  The paper only reports the success rate of the methods on the WebArena-Lite benchmark. Additional metrics, such as the average number of steps required to complete a task, would provide a more comprehensive assessment.
>
> Below are additional metrics, including average steps and a success/failure breakdown, as suggested. To provide a more comprehensive assessment, we have also provided a breakdown comparing successful and unsuccessful tasks across the different websites/tasks. Furthermore, we have run experiments on other datasets including WebVoyager. Please see R2-1 response below.
>
> | Website           | # Tasks | Avg. Steps (All) | Avg. Steps (Success) | Avg. Steps (Fail) | Success Rate (%) |
> | ----------------- | ------- | ---------------- | -------------------- | ----------------- | ---------------- |
> | Overall           | 165     | 11.12            | 7.52                 | 13.43             | 53.9%            |
> | GitLab            | 30      | 13.7             | 5.98                 | 20.35             | 53.3%            |
> | Reddit            | 19      | 9.37             | 8.31                 | 9.92              | 84.2%            |
> | Shopping Admin    | 35      | 12.4             | 8.65                 | 14.41             | 48.6%            |
> | Shopping          | 45      | 9.87             | 7.11                 | 10.66             | 55.6%            |
> | Map               | 26      | 10.00            | 10.37                | 9.10              | 46.2%            |
> | Multiple Websites | 10      | 11.70            | 6.00                 | 17.83             | 30.0%            |
>
>
>
> > R1-2: The proposed framework lacks novelty, as it essentially follows a hierarchical planning approach and utilizes environmental feedback for replanning—both of which are commonly used in planning applications.
>
> We would like to direct the reviewer to our response R3-4 in our response to Reviewer 3.
>
>
> > R1-3: The experiments are conducted solely on WebArena-Lite, a simulated environment. It would be more informative to evaluate the approach on WebVoyager, which better reflects real-world web behavior.
>
> We conducted new experiments on WebVoyager, please see R2-1 in our response to Reviewer 2, where Plan-and-Act achieves SOTA results for text-only models on WebVoyager with an accuracy of **80.02%**.
>
>
> > R1-4: More prior methods [1, 2] should be included in the experimental comparison to better contextualize the improvements.
>
> AgentOccam [1] and WebPilot [2] do not report results on WebArena-lite, so we evaluated our method on the full WebArena benchmark; please see R2-2 for a detailed comparison, where we find that Plan-and-Act achieves performance on-par or better with prior work on WebArena.
>
>
> > R1-5: The writing could be improved in the following aspects: (1) The Related Work section should not merely summarize previous studies but should also explicitly discuss the similarities and differences between the proposed approach and existing methods. (2) The paper should include a thorough discussion of the limitations of the proposed method.
>
> Regarding related work, we will expand it based on the discussion in R3-4.
>
> Regarding the limitations, one main drawback is that Action Trajectory Generation (Section 4.1) does depend on having a baseline model that can successfully complete the web tasks. The synthetic data generation pipeline introduced in Section 4.3 is able to mitigate some of these concerns such that with a sufficient amount of training data. However, for datasets that do not have any training data, such as WebVoyager, the pipeline will depend on having a base model to collect trajectories.
>
> We will include a more thorough discussion of limitations in the final version of the paper.

---

> > ### Comment · Reviewer_4dS1 · 2025-04-08
> >
> > Thanks for the authors' comprehensive response. It addressed my concerns. So I increased my score.

---

### Decision · Program_Chairs · 2025-05-01

**Decision:**

Accept (poster)

**Comment:**

The paper studies LLM planning with separate planner and executors. The idea is relatively simple and has been applied in prior work, and this looks like the main concern from the reviewers. The authors partially address such concerns, and as the result two reviewers raised their scores to 3, and right now we have 3, 3, 3, 3 as the final scores. By reading the comments and rebuttals and relevant parts in the paper, I can understand that the main contribution of the paper seems to be using synthetic data to train the planner, while most prior studies simply use existing LLMs with prompting. There are also studies using synthetic data, but the authors claim that prior studies rely on simulators while their approach is simpler. Overall, my assessment on the paper is that the paper's contribution is to combine several relatively simple ideas (separate planner and executor, synthetic data for training, data generation using LLM instead of simulator, etc.) and provide a concrete implementation and evaluation. The research direction is very important and the authors provide a reasonable study to investigate a valid design point in this direction. Thus I would like to recommend weak accept (accept if there is room in the program) to the paper.